# Molecular landscape and functional characterization of centrosome amplification in ovarian cancer

Carolin M. Sauer [1,2] ✉, James A. Hall [1,2], Dominique-Laurent Couturier [1,3], Thomas Bradley [1,2], Anna M. Piskorz[1,2], Jacob Griffiths[1,2], Ashley Sawle [1,2], Matthew D. Eldridge [1,2], Philip Smith [1,2], Karen Hosking[2], Marika A. V. Reinius [1,2,4], Lena Morrill Gavarró [1,2], Anne-Marie Mes-Masson [5], Darren Ennis[6], David Millan[7], Aoisha Hoyle [8], Iain A. McNeish [6], Mercedes Jimenez-Linan[4], Filipe Correia Martins [1,2,4], Julia Tischer[1,2], Maria Vias[1,2] & James D. Brenton [1,2,4] ✉

High-grade serous ovarian carcinoma (HGSOC) is characterised by poor outcome and extreme chromosome instability (CIN). Therapies targeting centrosome amplification (CA), a key mediator of chromosome missegregation, may have significant clinical utility in HGSOC. However, the prevalence of CA in HGSOC, its relationship to genomic biomarkers of CIN and its potential impact on therapeutic response have not been defined. Using high-throughput multi-regional microscopy on 287 clinical HGSOC tissues and 73 cell lines models, here we show that CA through centriole overduplication is a highly recurrent and heterogeneous feature of HGSOC and strongly associated with CIN and genome subclonality. Cell-based studies showed that high-prevalence CA is phenocopied in ovarian cancer cell lines, and that high CA is associated with increased multi-treatment resistance; most notably to paclitaxel, the commonest treatment used in HGSOC. CA in HGSOC may therefore present a potential driver of tumour evolution and a powerful biomarker for response to standard-of-care treatment.

High-grade serous ovarian carcinoma (HGSOC) accounts for most ovarian cancer cases related deaths with a five-year survival rate of less than 30%. Overall survival of HGSOC patients has not changed over the last two decades mainly due to the severe ongoing chromosomal instability (CIN) characterising and driving this disease. CIN is present in virtually all HGSOC[1], drives resistance to anticancer therapies[2–6], and is the main cause of regenerating subclonal diversity in response to treatment-induced selective pressures[7]. The lack of a clear understanding of the drivers of this genomic complexity in HGSOC has significantly impeded the development of precision therapies, with the major exception being the successful targeting of BRCA1/2 dysfunction with PARP inhibitors. Patterns of copy number aberrations in

[1]Cancer Research UK Cambridge Institute, University of Cambridge, Li Ka Shing Centre, Robinson Way, Cambridge CB2 0RE, UK. [2]Cancer Research UK Major Centre–Cambridge, University of Cambridge, Cambridge CB2 0RE, UK. [3]Medical Research Council Biostatistics Unit, University of Cambridge, Cambridge CB2 0SR, UK. [4]Cambridge University Hospital NHS Foundation Trust and National Institute for Health Research Cambridge Biomedical Research Centre, Addenbrooke's Hospital, Cambridge, UK. [5]Department of Medicine, Université de Montréal and Centre de recherche du Centre hospitalier de l'Université de Montréal (CRCHUM), Montreal, QC, Canada. [6]Department of Surgery and Cancer, Ovarian Cancer Action Research Centre, Imperial College London, London, UK. [7]Department of Pathology, Queen Elizabeth University Hospital, Glasgow, UK. [8]Department of Pathology, University Hospital Monklands. NHS Lanarkshire, Airdrie, UK. ✉e-mail: carolin.sauer@cruk.cam.ac.uk; james.brenton@cruk.cam.ac.uk

HGSOC can identify specific mutational processes causing CIN and highlight potential therapeutic vulnerabilities[8,9], but the relationship between these signatures and other cellular causes of CIN, specifically centrosome amplification has not been investigated.

The single most prominent cause of CIN is chromosome missegregation, which can be caused by weakened mitotic checkpoint, improper chromosome attachment to the mitotic spindle and centrosome amplification (CA; typically defined as an abnormal number of centrosomes >1 in nondividing cells)[10–12]. The centrosome, also known as the microtubule organising centre (MTOC), consists of two centrioles embedded within the pericentriolar material (PCM; a proteinaceous scaffold with microtubule-nucleating activities[13,14]), and guides spindle formation and accurate chromosome segregation during cell division[15]. Consequently, centrosome abnormalities, both structural and numerical, can lead to the missegregation of chromosomes, resulting in aneuploidy and CIN[10,16,17]. Centrosome abnormalities, such as CA, may provide a novel therapeutic target in cancers, and indeed, several drugs targeting centrosome duplication[18] or associated survival mechanisms[19,20] are now in development or clinical trials[21]. However detailed studies of the prevalence, mechanisms, and origins of CA in HGSOC and other cancers are now required for the clinical exploitation of this phenotype.

Most studies that have investigated the mechanisms and consequences of CA have experimentally induced or inhibited CA in cell lines or other model systems predominantly in breast cancer models. However, the high degree of variability in centrosome numbers observed across cancer cell populations indicates the existence of a CA "set point" or equilibrium[22] and suggests that cell lines have different tolerance thresholds for CA and maintain centrosome numbers through an equilibrium of CA mechanisms and negative selection[23,24]. Consequently, the experimental induction or inhibition of CA may trigger a range of different cellular responses that would not otherwise be observed without these artificial perturbations. In addition, because of significant challenges associated with detecting centrosomes in clinical human tissue samples, very few studies have investigated CA in clinical tumour specimens[25,26], and the prevalence of CA in HGSOC (both in cell lines and tumour tissues) remains largely unclear.

To address these shortcomings, we here describe a systematic and detailed analysis of baseline CA along with its molecular and genomic associations in large cohorts of HGSOC tissues and ovarian cancer cell lines. We develop a high-throughput microscopy-based assay to reliably detect and quantify centrosomes in formalin-fixed paraffin-embedded (FFPE) tissues and analyse centrosome profiles in >300 tissue samples containing 287 HGSOC tumours. These methods uncover a high prevalence of CA in HGSOC with marked intratumoural tissue heterogeneity. To further probe the biological nature of CA in ovarian cancer, we extend our approaches to the in-depth profiling of 73 ovarian cancer cell line models at single-cell resolution, and provide a comprehensive phenotypic, transcriptomic, and genomic characterisation of supernumerary centrosomes. The results of this work confirm the high prevalence of CA arising from centriole overduplication in ovarian cancer, directly link CA to CIN and genomic subclonality, and show that CA dictates treatment response to anti-mitotic agents, most importantly paclitaxel, with clear clinical implications. Finally, our work paves the way for future studies characterising centrosome abnormalities in large cohorts of clinical tumour specimens and provides an important resource for future research investigating CA and associated vulnerabilities for the treatment of HGSOC.

## Results

### Supernumerary centrosomes are present in the majority of HGSOC tissues and show marked inter- and intra-tumour heterogeneity

To investigate the presence and extent of supernumerary centrosomes in HGSOC, we selected full-face FFPE tumour sections ($n = 93$) from the prospective non-interventional cohort study (OV04) and tissue microarrays (TMAs; $n = 194$) from the British Translational Research Ovarian Cancer Collaborative (BriTROC) study (total of $n = 287$ tumour tissues; see Methods and Supplementary Table 1). We included matched and unmatched normal fallopian tube (FT) and other normal tissues as negative controls. Liver samples were included as a positive control for CA, since the liver is the only organ that has supernumerary centrosomes in normal cells[27–29]. To obtain accurate estimates of centrosome numbers in clinical HGSOC tissue samples, we developed an automated, high-throughput immunofluorescent microscopy-based imaging approach using the confocal Operetta CLS™ imaging system (Fig. 1a-b and Supplementary Methods). In total we profiled 3,632,389 nuclei across 11,811 imaging fields using up to 50 non-overlapping randomly placed imaging fields per tissue sample in 25 μm FFPE tissue sections. To allow comparisons across both cohorts and to account for potential batch effects, CA scores were normalised to the median CA score of normal control tissues within each cohort.

In both study cohorts, CA scores were significantly higher in tumour tissues than normal control tissues (FT, spleen and kidney; Kruskal-Wallis, $p \ll 0.001$). As expected, control liver tissues showed the highest CA score (Fig. 1d). Using a CA threshold based on the 95% confidence interval of the FT sample with the highest CA score (1.83; Fig. 2), 73% of OV04 and 59% of BriTROC tumour samples (63.5% combined) displayed significant CA (Fig. 1d). Only one tumour sample showed significantly lower CA scores than normal tissues (Fig. 2). Centrosomes in tumour tissues also had significantly larger PCM areas (measured as width × length determined from 2D maximum intensity projection images) than centrosomes detected in normal control tissues (Kruskal-Wallis, $p \ll 0.001$; Fig. 1e), indicating that structural centrosome abnormalities might also be a common feature in HGSOC.

As expected, we observed marked inter-tumoural heterogeneity for CA, but also substantial intra-tumoural differences within cases. Notably, there was marked variability in the distribution of CA scores across imaging fields, even between cases with similar median CA scores (Fig. 2). To investigate this further, we developed a hierarchical linear mixed model accounting for intra-tissue dependence of mean and variance CA scores (see Methods for details). Model estimates confirmed the significant differences in average CA levels between FT and tumour tissues ($p \ll 0.001$), as well as the presence of marked intratumoural heterogeneity ($p \ll 0.001$). To quantify the observed heterogeneity in CA phenotypes, we estimated the standard deviation of log-transformed CA scores across all imaging fields for each tissue sample. Examples of CA high and low tissues with varying tissue heterogeneity scores are shown as spatial heatmaps in Fig. 3a–b and Supplementary Fig. 1. Heterogeneity scores were significantly higher in OV04 samples (full face sections) than BriTROC samples (TMA cores; Wilcoxon, $p \ll 0.001$) consistent with a much larger area sampled by random imaging fields across full face sections compared to adjacent imaging fields covering 1 mm TMA cores. We also observed significant differences in CA levels between tumour tissues collected from the same patient ($n = 36$; Fig. 3c). Using the CA cutoff of 1.83, for the majority of patients (64%), analysed tissues fell into the same CA category (i.e. either significant CA or no CA), whereas for 36% of cases, tissues within either category were found. Importantly, no differences in CA scores were observed between different anatomical tissue sites (Kruskal-Wallis, $p = 0.62$; Fig. 3d), while tissue samples collected from the omentum showed higher CA tissue heterogeneity (Kruskal-Wallis, $p \approx 0.005$; Fig. 3e). However, this variability could result from differences in how blocks were sampled from pelvic and omental tissues between the two study cohorts.

Using robust statistical modelling, these analyses confirm very high prevalence of CA in HGSOC cases and significant intratumoural and intertumoural heterogeneity of CA phenotypes.

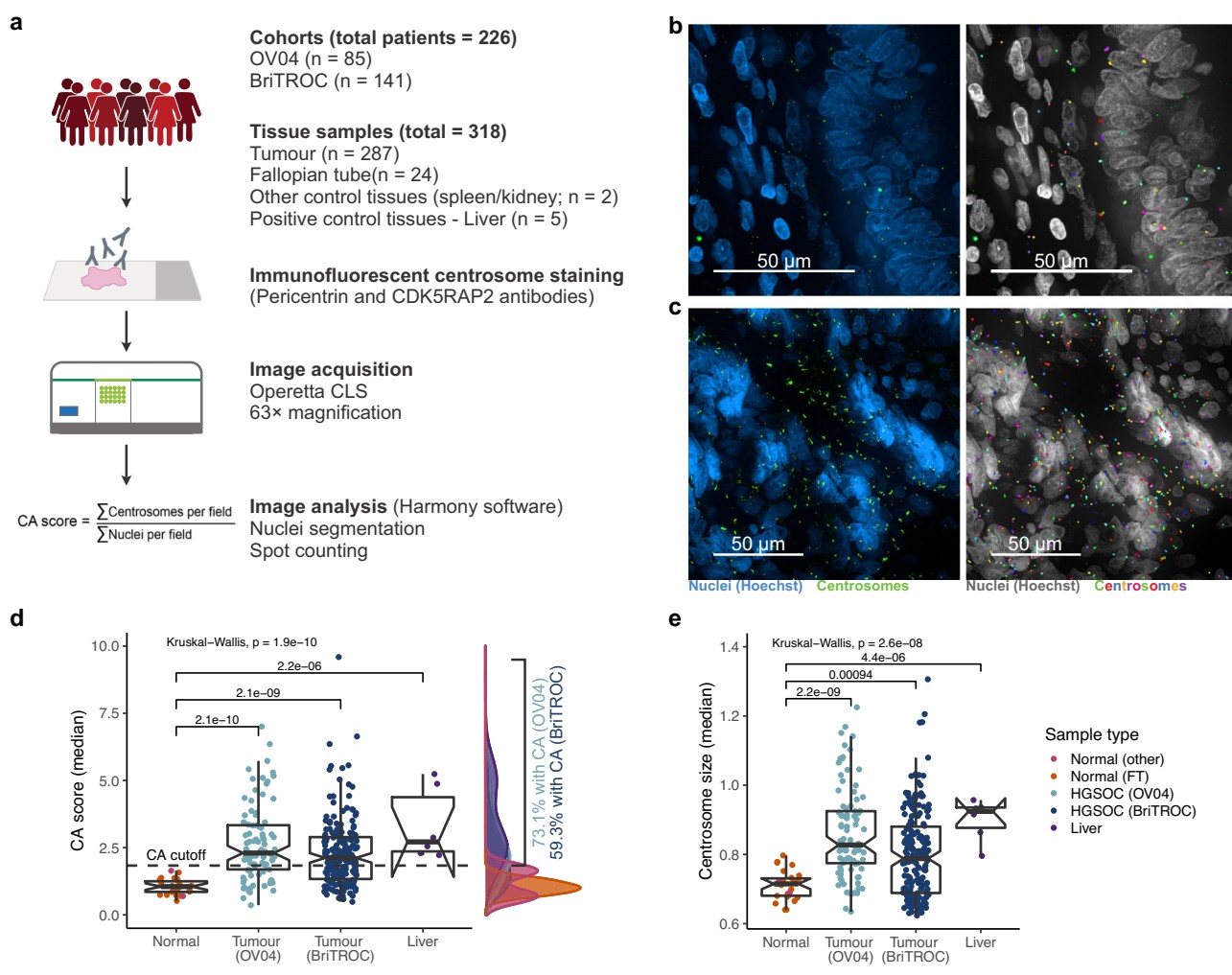

**Fig. 1 | Centrosome characterisation in HGSOC tissue samples. a** Basic workflow of centrosome characterisation in clinical tissue samples (also see Supplementary Methods). Example confocal immunofluorescent images (max. intensity projections) of a fallopian tube and HGSOC tissue are shown in (**b**) and (**c**), respectively (left panel). Nuclei are shown in blue, centrosomes are shown in green. Individually detected centrosomes following automated image analysis are highlighted in rainbow colours (right panel; nuclei shown in grey). Scale bars = 50 μm. **d, e** show sample type comparison of centrosome amplification (CA) scores and centrosome size, respectively. Sample types are indicated by different colours: Normal (other),

pink, $n = 2$; Normal (FT), orange, $n = 24$; HGSOC (OV04), light blue, $n = 93$; HGSOC (BriTROC), dark blue, $n = 194$; Liver, purple, $n = 5$. Statistics shown is a Kruskal–Wallis one-way analysis of variance test. Boxplots show 25th, 50th and 75th centiles; whiskers indicate 75th centile plus 1.5 × inter-quartile range and 25th centile less 1.5 × inter-quartile range. Notches on boxes extend 1.58 × inter-quartile range/sqrt($n$) approximating to the 95% confidence interval for comparing medians. Dashed horizontal line indicates CA cutoff threshold of -1.83. Source data are provided as a Source Data file.

## Centrosome amplification is not a prognostic marker in HGSOC

CA has been frequently associated with increased disease aggressiveness and poor patient prognosis in various cancer types[25,30]. We therefore investigated whether CA was associated with clinical or molecular features and overall survival in ovarian cancer. For patients with multiple tissue samples, the median CA score across these tissues was used. The BriTROC cohort, contained four patients with endometrioid ovarian cancer, which has improved prognosis compared to HGSOC patients. These patients showed significantly lower CA compared to the remaining HGSOC patients from both cohorts (Wilcoxon, $p = 0.02$; Fig. 4a). No significant differences in CA scores were observed across other important clinical variables including tumour stage (Kruskal-Wallis, $p = 0.19$), *BRCA1* and *BRCA2* germline mutation status (Kruskal-Wallis, $p = 0.36$), or between treatment-naïve (immediate primary surgery [IPS]) and post-chemotherapy initiation specimens (delayed primary surgery [DPS]; Wilcoxon, $p = 0.26$; Fig. 4b–d). We did, however, find an increase in CA tissue heterogeneity scores in DPS compared to IPS samples (Wilcoxon, $p = 0.015$; Fig. 4e) as well as an increase in centrosome

size (Wilcoxon, $p \approx 0.001$; Fig. 4f), suggesting that chemotherapy might influence tissue-wide CA characteristics.

The BriTROC study enrolled women who had relapsed following front-line therapy and who were well enough to undergo surgery or an image-guided biopsy[31]. As a result, the BriTROC cohort is enriched for better-outcome patients with a median survival of 4.9 years as compared to the median survival of 2.4 years in the OV04 cohort (Supplementary Table 1). We therefore performed survival analyses independently on both cohorts. Multivariable Cox proportional hazards showed that the presence of CA was not associated with overall survival in either of the two cohorts (OV04: $p = 0.461$, HR = 0.8, 95% CI 0.43–1.5; BriTROC: $p = 0.09$, HR = 1.59, 95% CI 0.93–2.7; Fig. 4g–h). This result was surprising given previous publications reporting CA as a poor prognostic variable in other epithelial cancers[25,30]. We further analysed the relationship between CA and survival using data from the Cancer Genome Atlas (TCGA; $n = 9,721$), using the CA20 gene expression signature as a proxy marker for CA[30,32]. This confirmed CA was highly prevalent in ovarian cancer which was amongst the highest CA20 expressing cancer type[30] (Supplementary

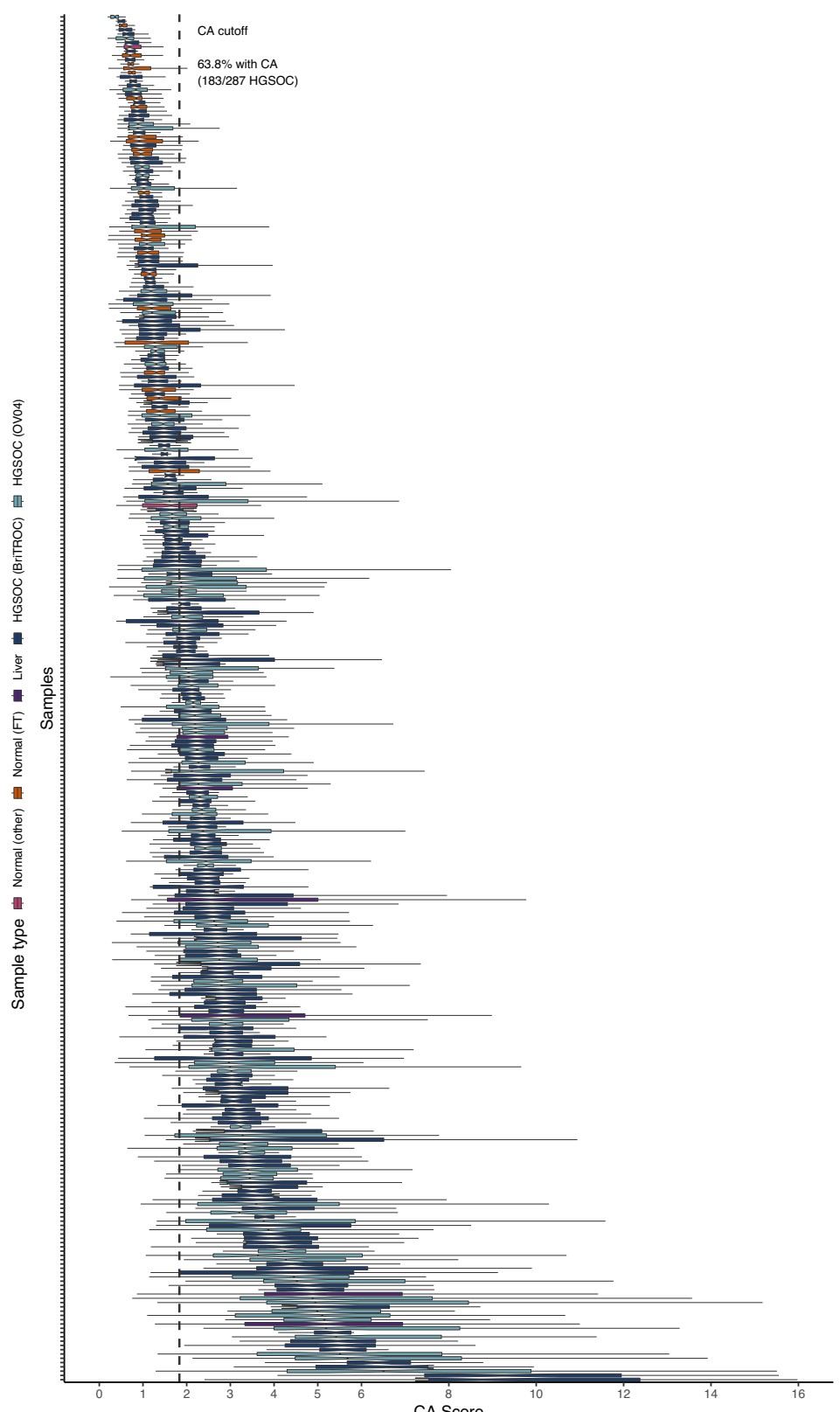

**Fig. 2 | Centrosome amplification profiles across 318 tissue samples.** Centrosome amplification scores across 287 HGSOC tissue samples, 24 fallopian tube, 1 spleen, 1 kidney, and 5 liver samples. Each boxplot represents CA scores across up to 50 imaging fields from individual tissue samples. Different tissue types are indicated by different colours. Dashed vertical line indicates the CA cutoff threshold of -1.83. Samples with median CA scores > CA cutoff were considered to have significant CA. Boxplots show 25th, 50th and 75th centiles; whiskers indicate 75th centile plus 1.5 × inter-quartile range and 25th centile less 1.5 × inter-quartile range. Notches on boxes extend 1.58 × inter-quartile range/sqrt(*n*) approximating to the 95% confidence interval for comparing medians. Source data are provided as a Source Data file.

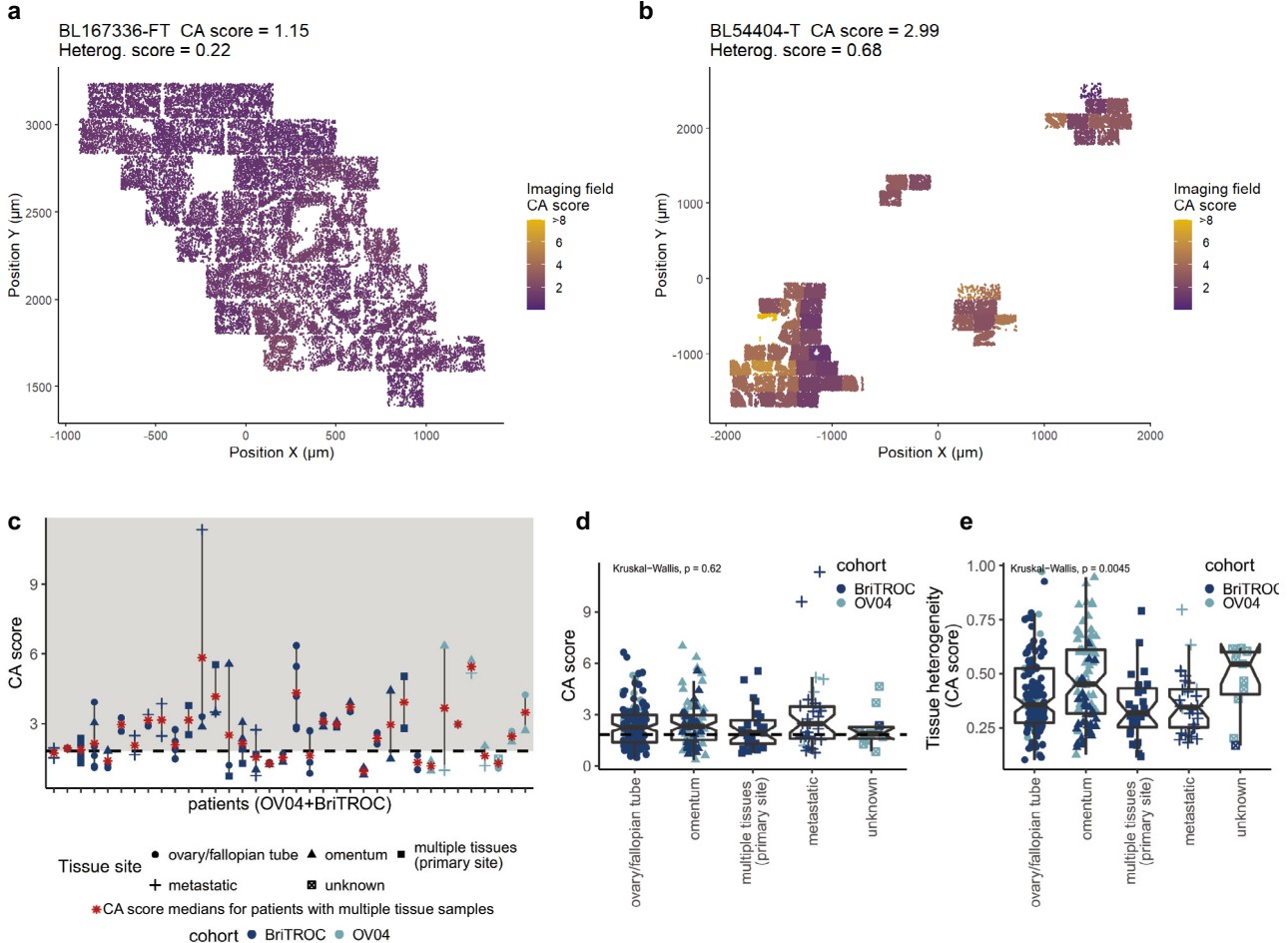

**Fig. 3 | Centrosome amplification shows inter- and intra-tissue heterogeneity.**
Spatial heatmap of centrosome amplification in a normal fallopian tube (**a**) and
HGSOC tumour tissue (**b**). Additional examples are shown in Supplementary Fig. 1.
CA scores are indicated by a colour gradient (purple = low, yellow = high) for each
imaging field. Points represent individual nuclei detected during image analyses
and were plotted in relation to their physical position (μm) on the microscope slide
(x and y axis). Note that CMYK printing may obscure differences on the CA score
colour scale. **c** Comparison of CA scores across tissues collected from individual
patients in a total of 36 cases. Red stars indicate CA score medians for patient with
multiple tissue samples. **d** Comparison of CA scores across different tissue sites

($n = 287$ individual samples; BriTROC, $n = 194$; OV04, $n = 93$). **e** Comparison of CA
tissue heterogeneity across different tissue sites ($n = 287$ individual samples; BriT-
ROC, $n = 194$; OV04, $n = 93$). Statistics shown for (**d**) and (**e**) is a Kruskal–Wallis one-
way analysis of variance test. Tissue sites are indicated by different shapes. Cohorts
are indicated by teal (OV04) and light blue (BriTROC) colours. Dashed horizontal
lines indicate the CA score cutoff of -1.83. Boxplots show 25th, 50th and 75th
centiles; whiskers indicate 75th centile plus 1.5 × inter-quartile range and 25th
centile less 1.5 × inter-quartile range. Notches on boxes extend 1.58 × inter-quartile
range/sqrt($n$) approximating to the 95% confidence interval for comparing med-
ians. Source data are provided as a Source Data file.

Fig. 2a). CA20 scores were strongly correlated with the CIN gene
expression signature, CIN25[33] (Spearman's R = 0.96, $p \ll 0.001$; Sup-
plementary Fig. 2b). This is consistent with previous studies reporting
that experimentally induced CA can give rise to aneuploidy and
CIN[10,34]. Importantly, we noticed that cancer types in which CA had
previously been implicated as a poor prognostic variable[25,30] (high-
lighted by blue arrowheads in Supplementary Fig. 2a) generally had
lower than the pan-cancer average for CA20 and CIN25 signature
expression (see Supplementary Fig. 2a). We hypothesised that the
survival differences observed for these cancers is caused by CA iden-
tifying cases with higher CIN. We further reasoned that in cancer
subtypes that are primarily driven by severe CIN, such as HGSOC[35],
there is less CA effect on universally poor disease outcome. We
therefore separated the TCGA cancer subtypes into CIN-high and CIN-
low cancers (using a median split). Multivariable Cox proportional
hazard analysis on the CIN-low group confirmed previous findings that
high CA20 was associated with poor survival ($p < 0.001$, HR = 1.5, 95%
CI 1.2–1.9; Supplementary Fig. 2c). However, this association was not
observed in CIN-high cancers ($p < 0.971$, HR = 1.0, 95% CI 0.89–1.1;
Supplementary Fig. 2d). In addition, high CA20 was also not associated

with disease outcome in the TCGA HGSOC group ($n = 304$, patients
divided by CA20 median; log-rank test, $p = 0.63$), confirming our
observations in the BriTROC and OV04 cohorts.

## In-depth characterisation of ovarian cancer cell lines shows frequent centrosome amplification through centriole overduplication
Our data confirms that CA is a highly prevalent and heterogenous
feature in HGSOC tumours, but several unanswered questions remain.
PCM proteins are highly dynamic and can form assemblies without
centrioles[36]. Our tissue-based CA assay was limited to PCM proteins
(PCNT and CDK5RAP2) owing to significant limitations of other anti-
bodies in FFPE processed samples. It is therefore unknown whether
supernumerary centrosomes observed in HGSOC specimens are
functional and intact, or might alternatively be acentriolar PCM foci
caused by PCM fragmentation. To investigate this possibility, we per-
formed in-depth characterisation in a large panel of ovarian cancer cell
lines by developing a semi-automated imaging approach, as sum-
marised in Fig. 5a. Antibody panels were designed to allow quantifi-
cation of centrosomes (PCNT, CETN3, and CEP164) and micronuclei

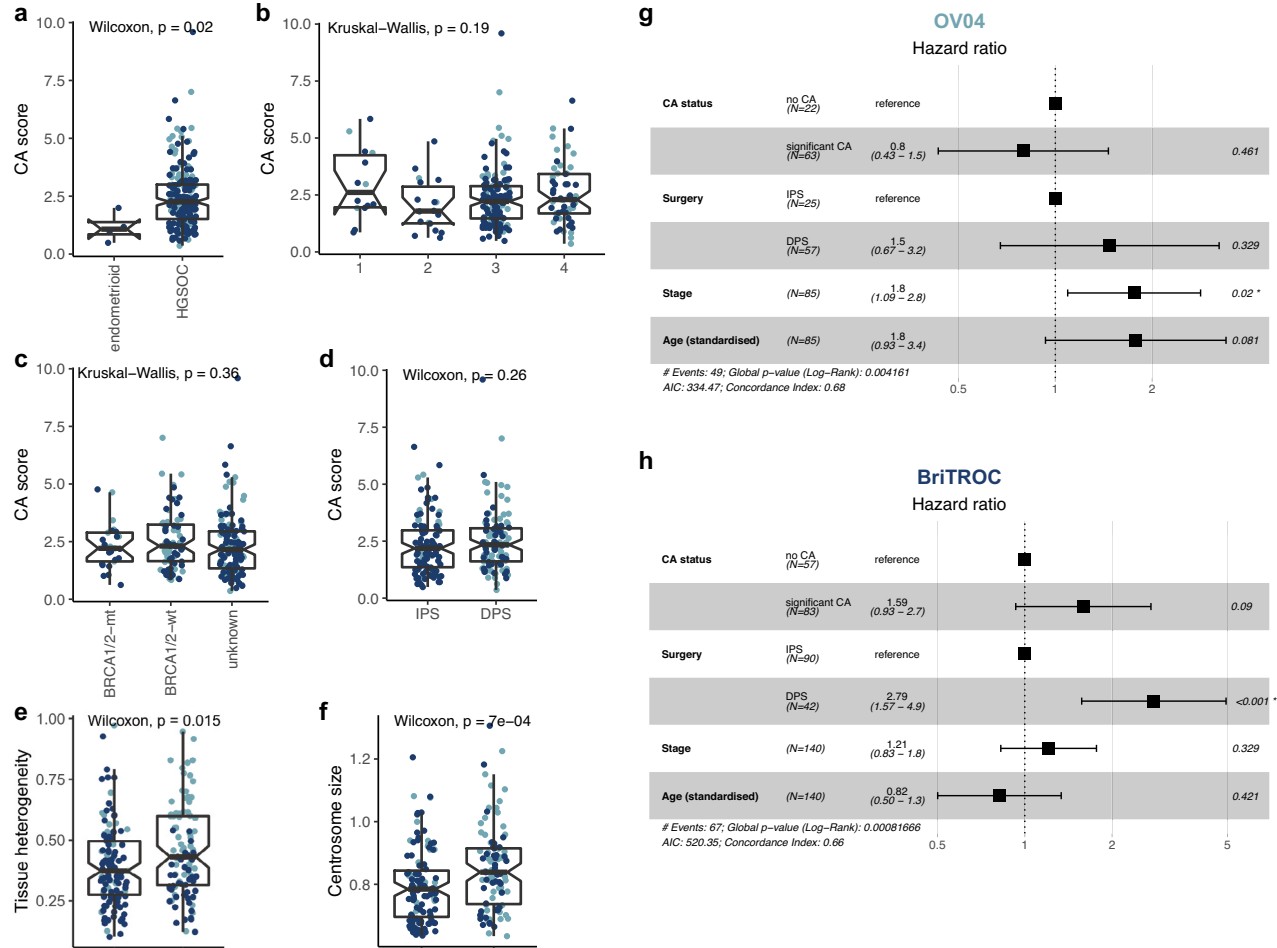

**Fig. 4 | Centrosome amplification is not associated with clinical features and disease outcome in HGSOC.** Comparison of CA scores across different (**a**) histo-types, (**b**) disease stages at diagnosis, (**c**) germline *BRCA1/BRCA2* mutation status, and (**d**) surgery types during sample collection (IPS = immediate primary surgery; DPS = delayed primary surgery). **e** Comparison of CA tissue heterogeneity in IPS vs. DPS cases. **f** Comparison of mean centrosome size in IPS vs. DPS cases. Boxplots show 25th, 50th and 75th centiles; whiskers indicate 75th centile plus 1.5 × inter-quartile range and 25th centile less 1.5 × inter-quartile range. Notches on boxes extend 1.58 × inter-quartile range/sqrt($n$) approximating to the 95% confidence interval for comparing medians. **a**,**d**–**f** show unpaired two-sided Wilcoxon tests.

**b**–**c** Statistics shown is a Kruskal–Wallis one-way analysis of variance test. **a**–**f** depict 287 individual tumour samples (BriTROC, $n$ = 194; OV04, $n$ = 93). OV04 samples are shown in light blue, BriTROC samples are shown in dark blue. Note that for patients with multiple tissue samples, the median CA score across these tissues was used. Source data are provided as a Source Data file. **g**–**h** Forest plots of multivariable Cox proportional hazard modelling on overall survival for OV04 and BriTROC patients respectively with and without CA. Adjusted covariates included surgery type, stage and age. Squares display the hazard ratio (HR) and whiskers the 95% confidence intervals of the HR. p-values shown are derived from likelihood ratio tests.

(Hoechst) in single cells (cytokeratin stains were used to estimate cell boundaries), as well as the estimation of mitotic indices (phospho-histone H3; pHH3), and the detection of DNA damage ($\gamma$H2AX). Combined, we analysed a total of 388,748 cells from 73 screened ovarian cancer cell lines (Fig. 5b). Cell lines showed a mean CA frequency of 26.2% (range 0.5–50.4%; defined as non-mitotic, pHH3-negative interphase cells with two or more centrosomes). These analyses confirmed that ovarian cancer cell lines recapitulate the high prevalence of CA observed in HGSOC tissue samples. In addition, the following cell lines exhibited very low numbers of detected centrosomes: PEO6, PEO16 and PEO4, suggesting likely centrosome loss. Example images of cell lines with low and high CA are shown in Fig. 6a, b.

To examine whether supernumerary centrosomes were intact centrosomes or aggregates of PCM, we analysed co-staining of PCNT with the centriole markers Centrin 3 (CETN3) and CEP164. CEP164 is a distal appendage marker, which can only be found on the mother centriole of mature centrosomes[37]. Accordingly, acentriolar PCM assemblies would be expected to be negative for both CEP164 and

CETN3 markers. Extra centrosomes caused by the incorrect or premature segregation of centrioles would be expected to result in CETN3-positive PCM foci with approximately 50% showing positive CEP164 staining. Lastly, CA arising from true centriole overduplication would be expected to result in fully functional centrosomes containing two centrioles, a daughter and a CEP164-positive mother centriole, surrounded by PCM. We found that both markers, CETN3 and CEP164, were present in the large majority (>95%) of detected PCM foci (Fig. 5b). This confirms that CA in ovarian cancer cells is predominantly caused by centriole overduplication resulting in intact and therefore likely fully functional centrosomes.

Given the high degree of CIN observed in HGSOC in comparison to other ovarian cancer subtypes, we next asked whether HGSOC cell lines showed higher levels of CA, and whether CA correlated with other molecular features of interest, including DNA content and the presence of micronuclei. We detected micronuclei in the majority of analysed cell lines with micronuclei frequencies ranging from 2.7–48.9% (mean 15.6%; defined as single nonmitotic, pHH3-negative interphase cells which contain at least one or more micronuclei). High

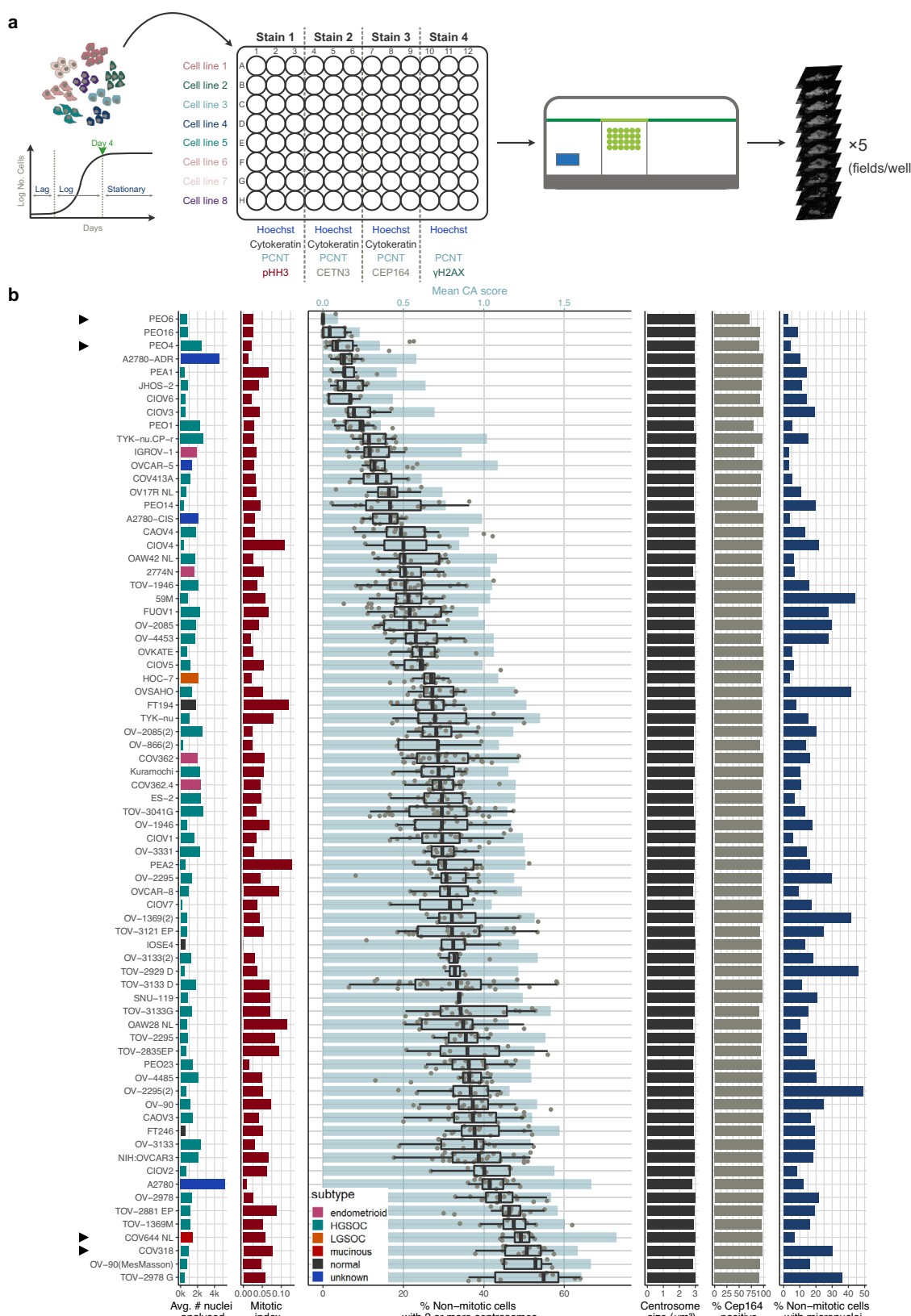

level CA was associated with increased MN frequencies (Spearman's $R = 0.42$, $p \ll 0.001$), providing further evidence that CA in ovarian cancer cell lines may induce MN formation through chromosome missegregation. In addition, we found moderate to strong correlations between CA frequencies and nuclei intensities (surrogate marker for DNA content; Spearman's $R = 0.32$, $p = 0.006$) as well as increased mitotic indices (Spearman's $R = 0.38$, $p < 0.001$). No correlation was observed between CA frequencies and DNA damage (measured as percentage of cells with γH2AX-positive nuclei Spearman's $R = 0.14$, $p = 0.3$; Fig. 6c). Importantly, we also did not observe significant differences in CA frequencies across different histological subtypes (Wilcoxon, $p = 0.17$; Fig. 6d) including the detection of mid- to high-

**Fig. 5 | In-depth imaging-based characterisation of ovarian cancer cell lines.**
**a** Cell line characterisation workflow. Growth curves were estimated for each cell line to determine optimal seeding densities. Cells were then seeded according to estimated seeding densities into an optical, poly-L-lysine coated 96 well plate and incubated for 4 days until they reached confluence (late Log/early Stationary phase). Cells were fixed with 100% methanol and stained against indicated proteins. For each well, five independent fields were imaged using the indicated Operetta CLS™ system (40× confocal water objective). **b** Immunofluorescent imaging results for 73 screened cell lines ordered by the median percentage of nonmitotic cells with 2 or more centrosomes (boxplots). Boxplots show 25th, 50th and 75th centiles; whiskers indicate 75th centile plus 1.5 × inter-quartile range and 25th centile less

1.5 × inter-quartile range. . Cell lines highlighted by grey arrowheads are depicted in Fig. 6a-b. Panel 1 shows the average number of nuclei included in each of the four staining screens. Bar plots are coloured by cell line subtype. Panel 2 indicates the mitotic index for each cell line. The fraction of non-mitotic cells with two or more centrosomes is shown in panel 3 (boxplots). Each point represents an individual imaging field. The mean (population wide) CA score ($= \frac{\sum Centrosomes}{\sum Nuclei}$) is shown by light blue bars. Panel 4 shows the centrosome size (estimated from max. intensity projection images), and panel 4 indicates the percentage of CEP164-positive centrosomes. Micronuclei (MN) frequencies, estimated as percentage of non-mitotic cells with at least one or more MN, are shown in panel 6. Source data are provided as a Source Data file.

level CA in the non-cancerous fallopian and ovarian surface epithelial cell lines, FT194, FT246 and IOSE4. These observations suggest that the transformation and/or immortalisation of normal cells, as well as selective pressures of culture conditions may induce oncogenic features that include CA. In line with these observations, we also observed notable copy number alterations in FT194 and FT246. In contrast, MN frequencies were significantly higher in HGSOC cell lines compared to all other histological cell line subtypes (Wilcoxon, $p \ll 0.001$; Fig. 6e). Nevertheless, our large-scale characterisation assay highlights CA through centriole overduplication as a highly prevalent feature and to be strongly correlated with the presence of MN in ovarian cancer cell lines.

### Reduction of oxygen levels induces CA in HGSOC cell lines
Hypoxia can promote CA via HIF1α-dependent induction of PLK4[38]. Out of the 73 cell lines screened, 25 were derived and maintained in low oxygen growth conditions (5% $O_2$; termed here as low-$O_2$ cell lines). Low-$O_2$ lines had significantly higher CA and MN frequencies to those grown in normoxia (21% $O_2$; Wilcoxon, $p < 0.001$ and $p \ll 0.001$, respectively; Fig. 6f, g) but also contained only HGSOC lines, which showed higher MN frequencies than lines from other ovarian cancer subtypes. To investigate whether increased CA and MN frequencies were caused by lower oxygen growth conditions, 8 HGSOC cell lines that are normally passaged in normoxia were transferred to 5% oxygen for 48 hours. Transient low oxygen growth conditions significantly increased CA (Wilcoxon, $p = 0.016$) but had no effect on MN frequencies (Wilcoxon, $p = 1$; Fig. 6h-i). Longer exposure to 5% oxygen conditions might be required to observe MN formation but these findings suggest that, at least in this experimental setting, CA precedes MN formation or MN formation is independent of CA.

### Cell lines with high centrosome amplification show upregulated survival signalling pathways
Having confirmed that HGSOC cell lines strongly phenocopied CA observed in patient samples, we next used RNA sequencing to investigate further potential drivers and consequences of CA. To understand possible confounding factors that might influence gene expression independent of CA and MN status, we first performed differential gene expression (DGE) analyses comparing different histological subtypes and cell culture growth conditions (see Supplementary Fig. 3). We did not observe significant upregulation of hypoxia hallmark/response genes in low-$O_2$ cell lines compared to cell lines cultured at 21% oxygen, and no additive effect was detected between CA and oxygen growth conditions when performing DGE model checks. Importantly, this suggests that it is the reduction of oxygen levels (compared to baseline) that induces CA (see Fig. 6h–i), and that the higher frequencies of CA and MN observed in low-$O_2$ cell lines are cell line intrinsic rather than caused by differing oxygen growth conditions. Based on these results, we developed an additive model that accounts for differences in histological subtypes to investigate differentially expressed genes in cell lines with high vs. low CA, and high vs. low MN frequencies (Fig. 7a, b). Consistent with the observation that CA and MN are moderately correlated

(Spearman's $R = 0.38$, $p \approx 0.002$; Fig. 6c), cell lines with either high CA or high MN frequencies showed similar gene expression signatures, including downregulation of E2F target genes, and upregulation of genes involved in interleukin and interferon signalling (Fig. 7c, d). The most upregulated pathway in MN-high cell lines was interferon alpha response signalling. In contrast, the most significantly upregulated pathway in CA-high cell lines was TNFα via NFκB signalling (Fig. 7c–f). NFκB/TNFα signalling pathways play pleiotropic roles in cell homeostasis and stress response[39,40], and might therefore be required for cell survival in response to CA.

### Centrosome amplification does not correlate with distinct patterns of copy number aberrations but is associated with increased genome subclonality
Multiple studies have reported putative associations between centrosome abnormalities and aneuploidy/CIN[10,30,34]. However, the underlying mechanisms for this association remain poorly understood. We hypothesised that supernumerary centrosomes might present a mutational process in HGSOC and induce distinct patterns of genomic aberrations. We therefore performed low-coverage/shallow whole genome sequencing (sWGS) and absolute copy number (ACN) fitting of HGSOC tissue samples to identify copy number signatures[8] that might be associated with the CA phenotype. No significant correlations were observed between copy number signatures and CA scores following $p$ value adjustment for multiple comparisons using samples with high confidence ACN fits ($n = 54$ OV04 and $n = 84$ BriTROC samples; Supplementary Fig. 4a, b) Further, no associations were found between CA and the number of breakpoints, copy number change points, oscillating copy number or copy number segment size (Supplementary Fig. 4c, d). Importantly, we also did not observe a correlation between CA scores and tumour ploidy (Supplementary Fig. 4e), suggesting that the high prevalence of CA in HGSOC is not causally related to whole genome duplication or cytokinesis failure.

ACN fitting from sWGS data from FFPE tumour tissue samples is challenging[41,42] and can be confounded by poor sequencing quality as a result of FFPE processing, tumour heterogeneity and low tumour purity. We therefore next used our cell line collection to further investigate potential CA-associated patterns of genomic aberration. Importantly, our ovarian cancer cell lines were highly representative of genomic features observed in the HGSOC cases (Supplementary Fig. 5)[8,43,44]. Consistent with our observations in HGSOC cases, we did not observe significant correlations between ploidy and CA (Spearman's $R = 0.16$, $p = 0.23$, respectively; Fig. 8a). This further confirms that cytokinesis failure and other processes impacting tumour ploidy may not be involved in driving CA in ovarian cancer.

To test the relationship between CA and CIN in our cell lines, we next quantified genome-wide copy number aberrations by estimating trimmed median absolute deviation from copy number neutrality (tMAD) scores[45] across all cell lines (Fig. 8b–d, Supplementary Fig. 5). Both CA and MN frequencies significantly correlated with tMAD scores (Spearman's $R = 0.36$, $p \approx 0.006$; Spearman's $R = 0.46$, $p < 0.001$, respectively); and cell lines which had high CA and high MN frequencies showed the most extreme genomic aberration (Spearman's

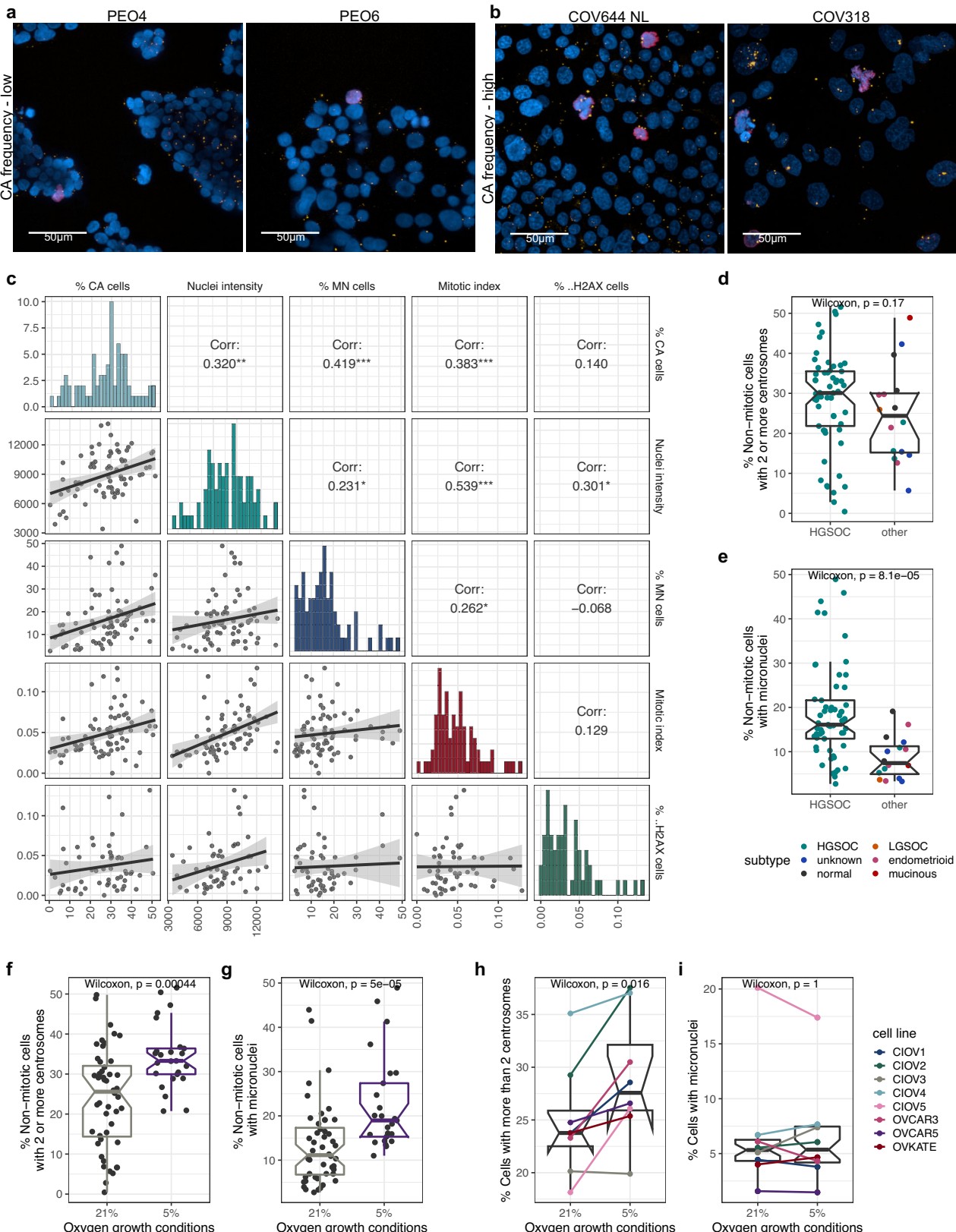

$R = 0.48$, $p < 0.001$; Fig. 8b–d). Consistent with our HGSOC cases, there were no significant associations between CA and copy number signatures (Fig. 8e). However, MN frequencies showed a significant negative correlation with signature 1 (Spearman's $R = -0.52$, $p \ll 0.001$, p.adj = 0.0003), and were positively correlated with signature 4 (Spearman's $R = 0.42$, $p \approx 0.001$, p.adj = 0.007), and signature 5

(Spearman's $R = 0.4$, $p \approx 0.002$, p.adj = 0.007; Fig. 8f). Given the compositional nature of copy number signatures, we also used a fixed effect model on the additive log ratio (ALR)-transformed copy number data to confirm these results and to test for global differential abundance of copy number signature exposures between both, CA high vs. low, and MN high vs. low cell lines (Fig. 8g, h). These analyses

**Fig. 6 | Centrosome amplification is associated with increased micronuclei frequencies and can be induced by low oxygen growth conditions.** Confocal immunofluorescent staining images (zoomed in from 40× max. intensity projection images) for (**a**) CA-low and (**b**) CA-high cell lines as highlighted in Fig. 5b. Cells were stained for Pericentrin (yellow), pHH3 (phospho-histone H3; mitotic cells; red), and DNA (Hoechst; blue). Scale bar = 50 μm. **c** Cell line characterisation correlation plot showing Spearman's rank correlations (two-sided) of analysed immunofluorescent staining results. P-values: *** $p < 0.001$; ** $p < 0.01$; * $p < 0.05$; . $p < 0.1$. Error bands show 95% confidence intervals. **d–e** Cell line subtype comparison of centrosome amplification and micronuclei frequencies, respectively ($n = 73$ individual cell lines; HGSOC, $n = 57$; other, $n = 16$). Histological subtypes are indicated by different colours. Note that the three "normal" cell lines were grouped together with other subtypes, as these cell lines are transformed and show some cancer characteristics.

**f–g** Centrosome amplification and micronuclei frequencies, respectively, in cell lines grown at 21% vs. 5% oxygen ($n = 73$ individual cell lines; 21% oxygen, $n = 48$; 5% oxygen, $n = 25$). **h–i** Cell lines, which are normally grown in normoxic conditions (21% oxygen), were transferred into 5% oxygen and centrosome amplification (**h**) and micronuclei frequencies (**i**) were estimated. Results depict unpaired two-sided Wilcoxon tests. Boxplots show 25th, 50th and 75th centiles; whiskers indicate 75th centile plus 1.5 × inter-quartile range and 25th centile less 1.5 × inter-quartile range. Notches on boxes extend 1.58 × inter-quartile range/sqrt($n$) approximating to the 95% confidence interval for comparing medians. Experiments were performed in four biological repeats for each cell line ($n = 8$), and four individual imaging fields were analysed for each repeat of each cell line. Data plotted shows the mean across all repeats. Source data are provided as a Source Data file.

confirmed no statistically significant shift in copy number signature activities between CA high and low cell lines (Wald test, p = 0.913), but showed a significant change in copy number signature abundances observed in MN high vs. low cell lines (Wald test, p ≈ 0.008). This shift is most likely caused by the decrease in signature 1 (corresponding to the signature with the most extreme ALR coefficient) in MN high cell lines. Copy number signature 1 is characterised by a low number of breakpoints and large copy number segment size[8]. Consistent with these results, MN frequencies were strongly correlated with a higher number of breakpoints per chromosome arm (Spearman's R = 0.52, $p \ll 0.001$) and negatively correlated with mean segment size (Spearman's R = −0.51, $p < 0.001$). In contrast, we did not observe any correlations between CA frequencies and genomic features, including segment size, number of breakpoints, copy number change points, and oscillating copy number (Supplementary Fig. 6).

Finally, ongoing CIN is considered a principal mediator of intra-tumour heterogeneity[11]. Given the strong link between CA and CIN, we estimated fractions of subclonal segments (genome subclonality) from sWGS data as previously described[41,46] for all sequenced cell lines. Strikingly, CA frequencies were significantly correlated with genome subclonality (Spearman's R = 0.34, $p \approx 0.009$), and a similar trend was observed for MN (Spearman's R = 0.29, $p = 0.027$; Fig. 8i–j). This suggests that while CA is not associated with specific patterns of genome aberrations in our experimental setting (as read out by sWGS), it may play an important role in driving intra-tumour heterogeneity and tumour evolution.

### Cell lines with high centrosome amplification show decreased chemosensitivity

Experimentally expanding the degree of aneuploidy in cells can increase their ability to adapt and develop resistance to chemotherapeutic agents[2,5,6] and in this study, CA showed strong correlations with both genome subclonality and CIN (Fig. 8b–d, i). We therefore hypothesised that CA might be an integrative and quantitative biomarker for predicting resistance to chemotherapy in ovarian cancer cell lines. To test the relationship between CA and drug sensitivity, we selected seven CA-high and seven CA-low cell lines using a panel of eleven drugs focusing on standard-of-care chemotherapeutics as well as drugs targeting centrosome clustering, the spindle assembly checkpoint (SAC) and different components of the centrosome duplication and cell cycle (Fig. 9a). To control for variations in cell proliferation/division rates and assay duration, we used the growth rate inhibition (GR) metric[47,48] and estimated both potency (GR50, i.e., the concentration of a drug at which cell proliferation is reduced by 50%) and efficacy (GRmax, i.e., the maximum effect of a drug at the highest concentration tested) for each therapeutic agent (an ideal drug would have high potency and high efficacy).

We observed that drugs targeting SAC components or components of the centrosome duplication/cell cycle were highly effective in all fourteen cell lines inducing cytotoxic responses (Fig. 9b). By contrast, CW069, which specifically impedes centrosome clustering by

inhibiting the microtubule motor protein HSET, was the least effective and only induced partial cytotoxicity in four out of the 14 lines. Paclitaxel, which is the standard of care treatment for both newly diagnosed and recurrent HGSOC, was the most potent drug. Strikingly, cell lines with high CA had decreased drug response compared to cell lines with low CA. This observation was most marked for paclitaxel treatment which induced cytotoxicity in low CA lines, but only partial cell growth inhibition in high CA lines (Spearman's R = 0.65, $p = 0.014$; Fig. 9c). Similarly, BI2536, Volasertib (both PLK1 inhibitors) and AZ3146 (Mps1 inhibitor) showed significantly lower potencies, while Oxaliplatin, and Barasertib (Aurora B inhibitor) showed significantly lower efficacies in CA-high compared to CA-low cell lines (Fig. 9c). These data confirm that CA predicts differential response to mitotic and centrosomal targeting agents and has clear relevance to current use of paclitaxel in the clinic.

## Discussion

Centrosome amplification is highly prevalent in many cancer types. However, the in-depth characterisation of centrosome abnormalities and associated molecular features has been limited to small cohorts of tissue samples and cell lines, and previous approaches have not addressed the challenge of significant intra-tumoural heterogeneity. We developed a high-throughput microscopy-based approach to facilitate robust detection of centrosomes and to ensure adequate tissue representation using >10,000 images across >300 FFPE tissues. We found CA to be a highly prevalent and heterogeneous feature of HGSOC. Interestingly, two recent studies have reported centrosome loss as another centrosomal abnormality. In these studies, centrosome loss was associated with aneuploidy and disease progression in prostate cancer[49], and observed as a common feature in regions of ovarian cancer tissues in frozen sections[26]. While we also observed possible regions of centrosome loss in our study, only one tissue sample showed significantly lower CA scores (tissue-wide) than those detected in normal control tissues. Given the high-throughput data-driven approach deployed in this study we did not further inspect individual tissue regions for loss of centrosomes. Moreover, apparent lower counts for centrosomes may also arise from experimental biases including partial volume effects from tissue sectioning. Therefore, more detailed single cell analyses will be needed to further understand CA heterogeneity and refine estimates of centrosome loss. Nevertheless, consistent with the previous study[26], we observed large variability in the spread of CA scores across all imaging fields within the same tissue, confirming that the CA phenotype displays marked inter- and intra-tissue heterogeneity in HGSOC tumours[22,26]. Both the high degree of CA heterogeneity and the confounding effect of stromal cells underscore the importance of acquiring multiple imaging fields per tumour sample and highlight a key strength of our imaging approach (see Supplementary Methods and Supplementary Fig. 7–13).

In contrast to previous observations in other cancer types[25,30], we did not observe a correlation between the CA phenotype and disease outcome. This might be explained by the high degree of severe CIN

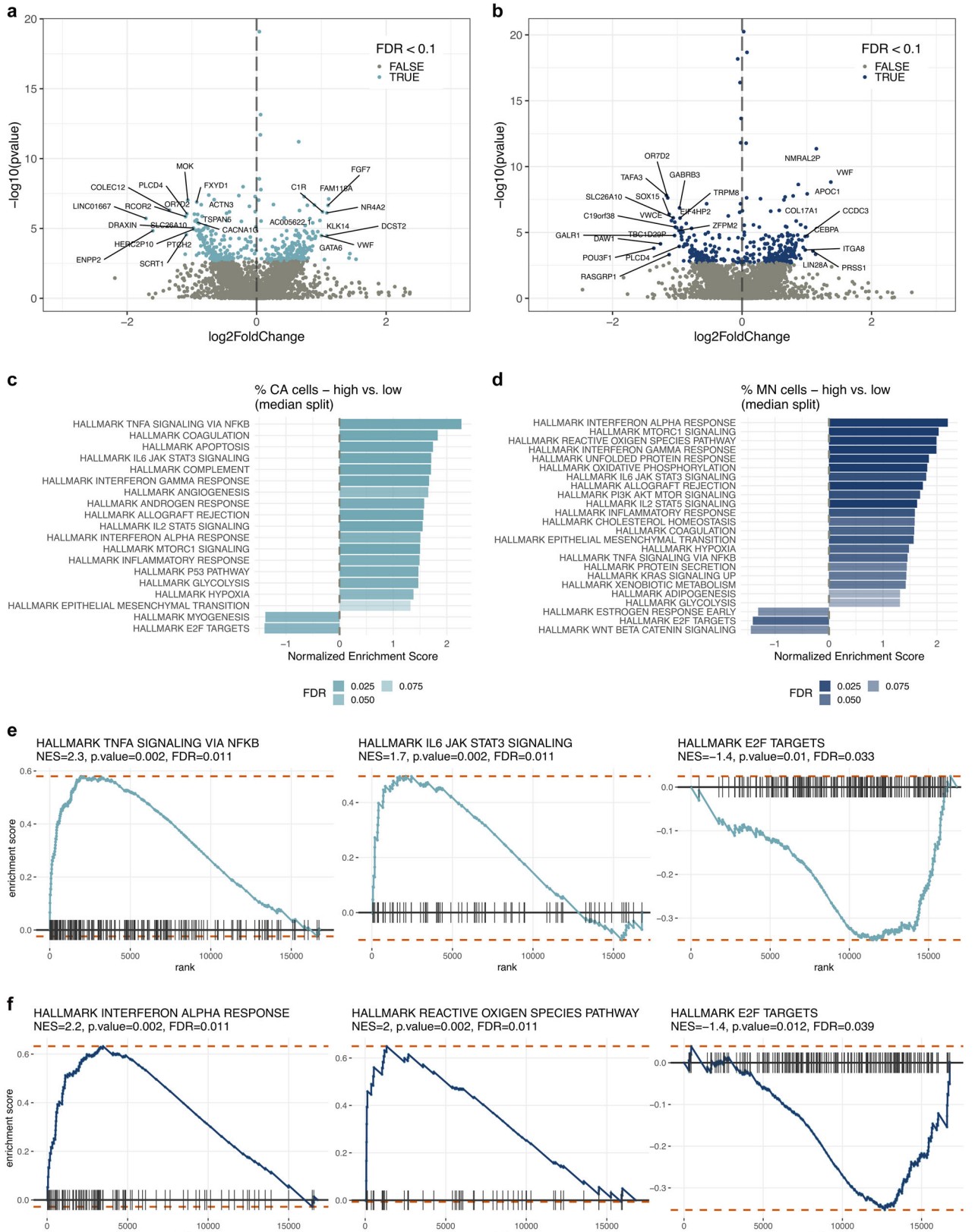

**Fig. 7 | Differential gene expression in cells with high centrosome amplification and micronuclei frequencies.** Differential gene expression volcano plots of all genes significantly regulated by (**a**) CA and (**b**) MN frequencies (high vs. low; median split). FDR < 0.1 highlighted in lightblue/blue. The top 25 most significantly up- or downregulated genes are labelled. **c,d** Gene set enrichment analysis (GSEA) of centrosome and micronuclei high vs. low cell lines, showing hallmark pathways for which *p* < 0.05 and FDR < 0.1. Examples of enrichment score plots for centrosome and micronuclei GSEA pathway results are shown in (**e**) and (**f**), respectively. Light blue/blue lines illustrate the running sum for each gene set shown. Maximum and minimum enrichment score (ES) are indicated by orange dotted lines. Leading-edge gene subset is indicated by vertical black lines. Source data are provided as a Source Data file. Gene set enrichment analysis and testing was performed using the fgsea R package.

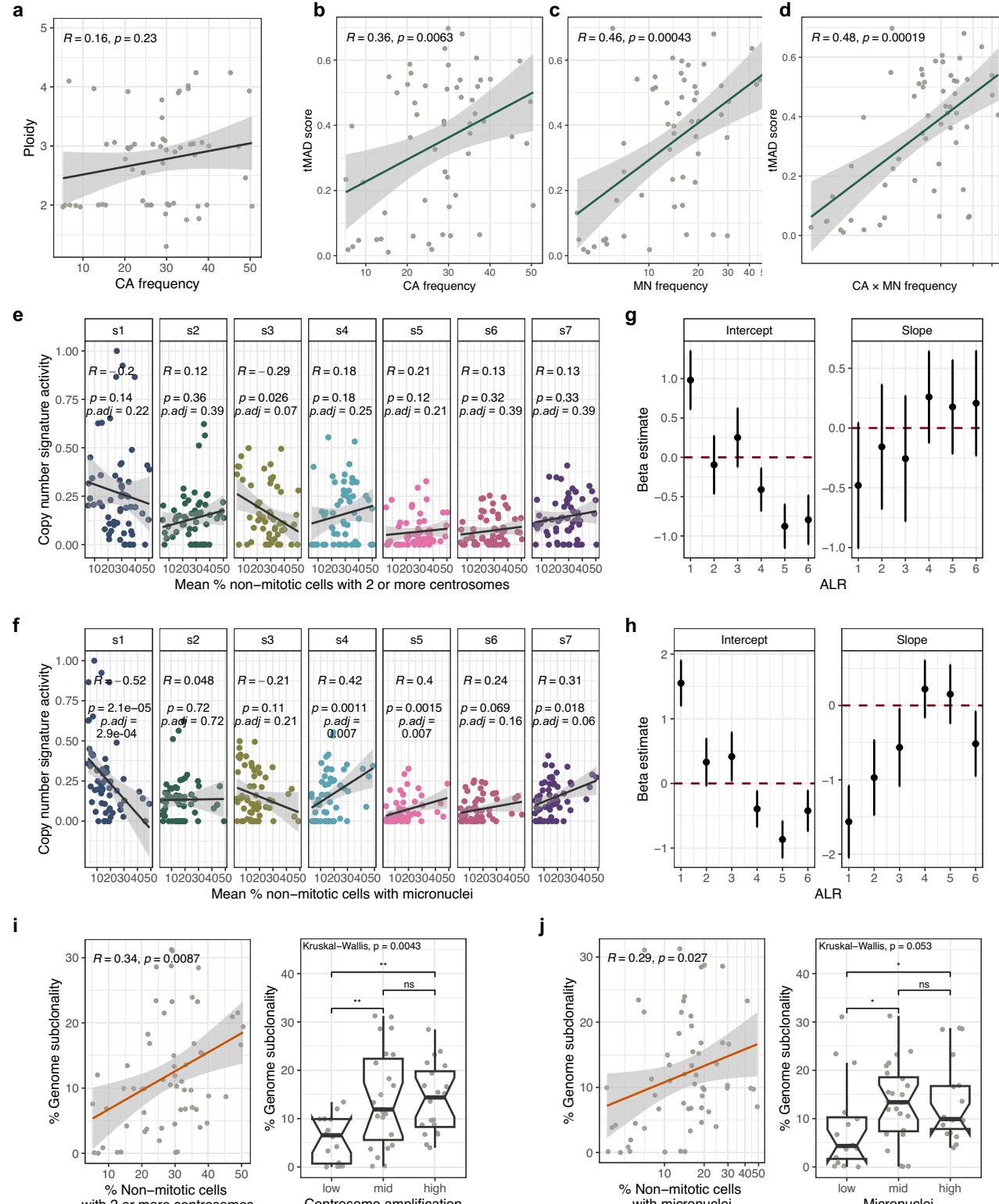

**Fig. 8 | Genomic characterisation of centrosome amplification and micronuclei. a** Correlation of centrosome amplification (CA) with cell line ploidy. **b**–**d** Correlations of cell line tMAD scores with CA frequencies, micronuclei (MN) frequencies and CA × MN (combined) frequencies, respectively. tMAD = trimmed median absolute deviation from copy number neutrality. Note that normal cell lines were excluded from this analysis. **e**–**f** Correlations of copy number signature exposures with CA and MN frequencies, respectively. Different signatures are indicated by colours. Intercepts and coefficients (slopes; depicted as dots) and their standard errors of the fixed effect model of ALR-transformed copy number signature data for CA and MN frequencies are shown in (**g**) and (**h**). **i**–**j** Correlations of genome subclonality estimated from ACN fits with CA and MN frequencies. Results are shown for $n = 59$ cell lines (for which matching sWGS and imaging data was available). All correlation coefficients were estimated using Spearman's rank correlations (two-sided). Boxplots show 25th, 50th and 75th centiles; whiskers indicate 75th centile plus 1.5 × inter-quartile range and 25th centile less 1.5 × inter-quartile range. Notches on boxes extend 1.58 × inter-quartile range/sqrt($n$) approximating to the 95% confidence interval for comparing medians. . Error bands show 95% confidence intervals. Source data are provided as a Source Data file.

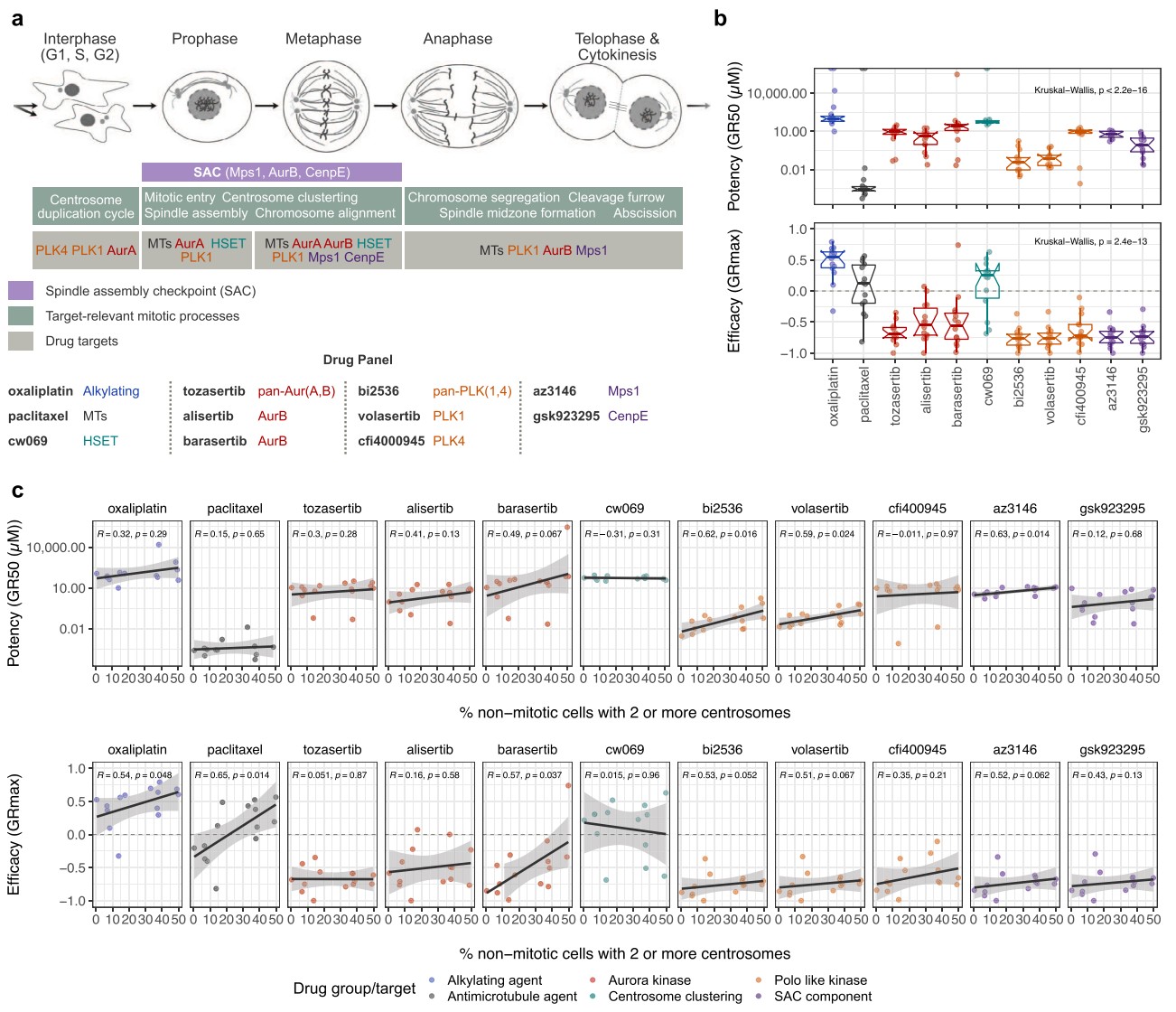

**Fig. 9 | Cell lines with high centrosome amplification show decreased drug sensitivities. a** Ovarian cancer cell line drug-screen overview showing drugs used in our drug sensitivity assay and their respective targets within the cell cycle. **b** Variation in potency and efficacy of selected drugs a cross all tested ovarian cancer cell lines (*n* = 14). Colours of boxplots indicate drug targets as shown in panel a. Boxplots show 25th, 50th and 75th centiles; whiskers indicate 75th centile plus 1.5 × inter-quartile range and 25th centile less 1.5 × inter-quartile range. Notches on boxes extend 1.58 × inter-quartile range/sqrt(*n*) approximating to the 95% confidence interval for comparing medians. **c** Correlation of centrosome amplification frequencies and drug potency (GR50; top panel) and drug efficacy (GRmax; bottom panel). Spearman's rank correlations (two-sided) are shown for each drug. Drug agents are colour-coded according to their respective drug target groups. Note that higher GR values correspond to lower potency or efficacy. The signs of GRmax values indicated drug response: GRmax > 0 indicates partial growth inhibition; GRmax = 0 indicates complete cytostasis; GRmax <0 indicates cell death (cytotoxicity). Error bands show 95% confidence intervals. Source data are provided as a Source Data file.

observed in virtually all HGSOC cases[1] which is itself associated with poor outcome.

In addition, we conclusively confirmed that CA through centriole overduplication is a widespread phenomenon in ovarian cancer cell lines, recapitulating the high prevalence of CA observed in HGSOC tissues. Cell lines with high CA were associated with the presence of micronuclei, increased nuclei intensity (often used as a surrogate for DNA content) and mitotic indices, suggesting that cells with CA might be delayed in mitosis. This is consistent with previous studies which have reported that cells with multiple centrosomes can take up to twice as long to complete mitosis, as additional time is required during the SAC to cluster centrosomes and align chromosomes in the metaphase plate[19,50–52]. However, these hypotheses need to be confirmed using time lapse microscopy to measure the length of mitosis in cells with and without CA.

HGSOC is characterised by frequent whole genome duplication and severe CIN. We recently reported the development of copy number signatures summarising distinct patterns of genomic features associated with CIN[8,9]. Notably, we did not observe correlations between CA and ploidy in either our HGSOC tissue sections or our ovarian cancer cell lines. This strongly suggests that while whole genome duplication through cell fusion, mitotic slippage and/or cytokinesis failure may induce CA in some cells, it is not the main driver of CA in ovarian cancer cell lines and tissues. In this study, CA did also not correlate with known HGSOC copy number signatures, in either tumour tissues or ovarian cancer cell lines. Importantly, early and ubiquitous *TP53* mutations in HGSOC provide a permissive environment for multiple mutational processes, including CA, to evolve and shape the HGSOC genome[8,12]. This makes it highly challenging to decipher distinct mutational patterns linked to CA. Single-cell

sequencing or long-read sequencing technologies might therefore be required to identify genomic patterns associated with CA. Nevertheless, our findings further support the notion that CA in HGSOC correlates with increased genomic aberration, can result in micronuclei formation, and thus likely presents an important contributor to CIN[10,16,53]. We also observed that high level CA is associated with increased genome subclonality. CA might therefore play an important role in driving intra-tumour heterogeneity and tumour evolution. Timelapse experiments will also be required to further investigate the fate of cells with amplified centrosomes, and to address the wider question of whether CA is merely a bystander or key driver of CIN. However, our data provides strong associations between CA and CIN/genomic subclonality, and while other mechanisms such as MN formation, mitotic checkpoint defects and replication stress have been described as important drivers of CIN[12], our study suggests that CA presents at least an important contributor to the genomic complexity observed in HGSOC.

Ongoing CIN and tumour heterogeneity are associated with increased treatment resistance by providing tumours with the ability to more readily adapt to environmental stresses[2,4–6,11]. In line with these findings, we observed decreased sensitivities to several therapeutic agents in cell lines with higher levels of CA. This observation was most marked in cells treated with paclitaxel. While paclitaxel induced cytotoxic responses in CA low cell lines, CA high cell lines only showed partial growth inhibition. Importantly, supernumerary centrosomes have increased capacities for microtubule nucleation[54]. Consequently, higher concentrations of paclitaxel might be required in cells with high CA to inhibit spindle assembly and thus cell division. We were underpowered to perform similar analysis in our clinical tissue sample cohort, as the majority of patients were treated with both carboplatin and paclitaxel, confounding the analysis of the predictive power of CA for paclitaxel response. In addition, no RECIST measurements for changes in tumour size were available to this study.

Centrosome-driven microtubule nucleation is regulated by the PCM, the size of which changes throughout the cell cycle and in preparation for spindle formation. We reported that HGSOC tumours frequently contain centrosomes with enlarged PCM foci in comparison to normal control tissues. However, due to FFPE staining limitations, we were not able to determine whether these size variations were true structural centrosome abnormalities or caused by staining artefacts and/or overlapping centrosomes that were not correctly segmented during spot counting. Further analyses on PCM size variations should be performed to investigate the role of enlarged PCM in CIN and paclitaxel sensitivity.

Lastly, RNA sequencing experiments revealed significant upregulation of TNFα/NFκB signalling in cell lines with high CA. Importantly, TNFα and NFκB signalling have been implicated in various aspects of oncogenesis[39,40,55,56]. Activation of NFκB results in the transcriptional upregulation of anti-apoptotic and antioxidant genes, promoting cell proliferation and survival. We therefore propose that NFκB signalling might be a key mediator of the CA "set point" allowing cells to tolerate higher levels of CA. Interestingly, a recent study illustrated CA-associated induction of the oxidative stress response via increased reactive oxygen species (ROS)[57]. High levels of ROS are toxic to cells and can induce cell senescence and apoptosis. Constitutive NFκB activity might therefore provide an important antioxidant mechanism resulting in cell survival and CA tolerance. NFκB-mediated upregulation of survival factors, together with increased levels of CIN and subclonality, might also contribute toward the CA-associated drug resistance observed in this study. In addition, NFκB signalling induces the transcription of many cytokines and chemokines, including IL6 and IL8, which regulate inflammatory responses, as well as cell invasion and migration[56]. Our RNA sequencing experiments showed that IL6 signalling was among the most upregulated signalling pathway in CA high cell lines. These findings further support the recent discovery

that CA induces non-cell autonomous invasion through the secretion of pro-invasive factors including both, IL6 and IL8[57]. Together, these data highlight NFκB as a potential mediator of CA-associated oncogenic features, such as increased resistance and invasion. Future studies will be needed to further investigate the exact mechanisms and relationship between NFκB signalling and CA in HGSOC.

In summary, our large-scale characterisation of centrosome profiles in both ovarian cancer cell lines and tissue samples revealed that CA, through centriole overduplication, is a highly prevalent yet heterogeneous feature of HGSOC, and links supernumerary centrosomes to CIN, genome subclonality and thus tumour evolution. In addition, we highlight the centrosome and CA-associated survival mechanisms as promising targets for novel therapeutic approaches in HGSOC granting further investigation. In particular, our data shows that CA predicts differential response to standard-of-care paclitaxel with clear implications for the clinical disease management of HGSOC. Finally, our work provides an important resource detailing the phenotypic, genomic, and transcriptomic characterisation of >70 ovarian cancer cell lines. This will facilitate the informed selection of suitable cell line models for future pre-clinical research investigating CIN, CA and associated vulnerabilities in ovarian cancer.

## Methods
We confirm that our research complies with all relevant ethical regulations. Please see section below on clinical samples and tissue processing for ethical approval information.

### Clinical samples and primary tissue processing and selection
HGSOC cases were selected from the CTCR-OV04 study, which is a prospective non-interventional cohort study approved by the local research ethics committee at Addenbrooke's Hospital, Cambridge, UK, (REC reference numbers: 07/Q0106/63; and NRES Committee East of England – Cambridge Central 03/018). All patients provided written informed consent. Tumour samples were processed following standardised operating protocols as outlined in the OV04 study design. For the generation of FFPE tissue blocks, tumour tissues were suspended in 10% neutral buffered formalin (NBF) for 24 hours and subsequently transferred into 70% ethanol for paraffin embedding and sectioning. Cases were chosen based on HGSOC histology and specimen availability from diagnosis, and represented real world clinical pathways with 29% from immediate primary surgery and 67% from delayed primary surgery. The sample set originally comprised a total of 120 FFPE tissues from 105 OV04 patients enrolled between January 2010 and April 2018. However, 27 samples were excluded prior to centrosome analyses owing to poor tissue quality and purity. The BriTROC-1 study enrolled patients with recurrent ovarian high-grade serous or grade 3 endometrioid carcinoma who had relapsed following at least one line of platinum-based chemotherapy between January 2013 and September 2017. Ethics/IRB approval was given by Cambridge Central Research Ethics Committee (REC reference number 12/EE/0349)[31,58]. All patients provided written informed consent. BriTROC TMAs were generated using 247 specimens collected at diagnosis from 172 cases. Three 1 mm cores were collected from each donor block and placed into three sister TMAs with identical layout. Prior to centrosome analyses, H&E stains of resulting TMAs were inspected, and tissue cores/samples that mostly contained stromal or necrotic regions were excluded ($n = 53$). An overview of all remaining tumour samples ($n = 287$) and patients ($n = 226$) from both cohorts included in this study is shown in Supplementary Table 1. In addition to the 287 tumour samples, we also included 24 normal fallopian tube (FT) samples, 1 normal spleen, 1 normal kidney and 5 normal liver samples.

### Cell lines
Cell lines used in this study are listed in Supplementary Table 2, together with their associated growth conditions and culture media. In

general, cell lines were grown at 37°C and 5% $CO_2$ according to ATCC/ECACC recommendations. OSE medium is composed of 50:50 medium 199 (Sigma-5017) and medium 105 (Sigma-6395). Cells were tested for mycoplasma contaminations on a regular basis using the qPCR PhoenixDx Mycoplasma kit (Procomcure Biotech), and cell line identities were confirmed prior to DNA extractions using our in-house human short tandem repeat (STR) profiling cell authentication service.

## Cell line growth curves

Cell line growth curves were generated to estimate optimal cell seeding densities for subsequent drug and centrosome characterisation assays. Cell lines were seeded into 96-well plates at different densities ranging from 5000–20,000 cells/well. Plates were placed inside a real-time live-cell system (IncuCyte®, Sartorius) and monitored for four days. Phase contrast images were acquired at 3-hour intervals and confluence measures were collected. Cell seeding densities that resulted in 80–90% confluency at the end of the assay (i.e. four days) were chosen for each cell line.

## Immunofluorescent centrosome staining

**FFPE tissue samples.** Tissue samples were fixed in 10% NBF for 24 hours, embedded in paraffin and cut into 25 µm thick sections. Subsequently, tissue slides were deparaffinised and rehydrated by resuspending them into 1×PBS three times for 5 min each. Antigen retrieval was performed via heat-induced epitope retrieval (HIER) in universal antigen retrieval buffer (ab208572, Abcam). Deparaffinised tissue slides were immersed in antigen retrieval buffer which was brought to boil using a conventional microwave. Once boiling temperature was reached, tissue slides were boiled for 10 mins at medium-high power and subsequently incubated at room temperature for 25–30 mins to allow buffer and slides to cool down.

Slides were washed three times in 1×PBS for 5 min each. Subsequently, tissue sections were permeabilised in permeabilisation buffer (5% BSA, 0.3% TritonX100) for 25–30 mins, and blocked in 5% BSA for 2 hours at room temperature.

Primary antibodies (anti-Pericentrin, Abcam ab28144; anti-CDK5RAP2, Gergely Lab) were diluted in 5% BSA, added to the tissue slides and incubated at 4 °C overnight. The anti-CDK5RAP2 antibody was used at a concentration of 2 µg/ml and the ab28144 anti-Pericentrin antibody was used at a concentration of 5 µg/ml. Following incubation with primary antibodies, tissue slides were thoroughly washed three times in 0.1% Tween 20 in 1×PBS for 8 mins each. Secondary antibodies (Alexa Fluor 488 and Alexa Fluor 555, cat No.: A-11001 and A-21429, Invitrogen) were diluted in 5% BSA to a concentration of 4 µg/ml (1:500) and incubated for 1 hour at 37 °C. Subsequently, slides were washed again three times in 0.1% Tween 20 in 1×PBS for 8 mins each, rinsed with distilled water and counterstained with 1 µg/ml Hoechst (33342, B2261 Sigma-Aldrich) for 15 mins at room temperature. Tissue slides were mounted in glycerol / n-propyl gallate medium (4% n-propyl gallate (02370, Sigma-Aldrich), 8 ml 100% glycerol, and 2 ml 1×PBS) and incubated for 24–48 hours protected from light to allow tissue equilibration and diffusion of mounting media prior to imaging. Please also refer to Supplementary Methods and Supplementary Figs. 7-13 for more information on method development.

**Cell lines.** Optical 96–well plates (CellCarrier-96 Ultra, Perkin Elmer) were coated with sterile Poly-L-Lysine (P4832, Sigma). Cells were seeded using previously estimated cell seeding densities and cultured for four days until confluent (see Fig. 5a for an overview of the experimental set-up). Cells were fixed in 100% ice-cold methanol for 5 mins at −20 °C, subsequently washed three times in 1×PBS and stored in 200 µl PBS at 4 °C until further use. Cells were rinsed with 0.1% Tween 20 in 1×PBS and permeabilised with 1% TritonX100 + 0.5% NP40 in 1×PBS for 5 mins at room temperature. Subsequently, cells were blocked for

1 hour at room temperature using 5% BSA in 1×PBS. Primary non-conjugated antibodies were diluted in blocking buffer, added to the cells and incubated overnight at 4 °C. An overview of antibodies, provider information, and concentrations for each of the four stains is shown in Supplementary Table 3. Following primary antibody incubation, cells were washed three times with 0.1% Tween 20 in 1×PBS. Secondary antibodies (Alexa Fluor 555 and Alexa Fluor 647, cat No.: A-21429 and A-21236, Invitrogen) were diluted to 1 µg/ml (1:2000) in blocking buffer, added to the cells and incubated for 1 hour at 37 °C. Subsequently, cells were washed three times with 0.1% Tween 20 in 1×PBS, and re-blocked with 5% BSA in 1×PBS for 30 mins at room temperature. Subsequently, conjugated primary antibodies were diluted, added to the cells, incubated for 1 hour at room temperature, after which cells were washed again three times with 0.1% Tween 20 in 1×PBS. After the last washing step, cells were rinsed in Milli-Q water. Hoechst 33342 was diluted to 1 µg/ml in MilliQ water, added to the cells and incubated for 15-30 mins at room temperature. Cells were washed and stored in 1×PBS until further use.

## Microscopy

**SP8.** For method optimisation purposes (see Supplementary Methods and Supplementary Fig. 7-9), sequential confocal images were acquired using a Leica TCS SP8 microscope. Z-stacks were collected at a step size of 0.35 µm. The pinhole diameter for channels containing centrosome staining signals was set to 0.5 AU. White light lasers with wavelengths of 488 and 555 nm were used to excite AlexaFluor 488 and AlexaFluor 555, respectively. Hoechst signal was excited using a laser diode 405.

**Operetta.** For large scale centrosome profiling experiments, the Operetta CLS™ high-content analysis system was used. For FFPE tissue sections, the Operetta PreciScan™ feature was used to allow automatisation of imaging processes—whole microscopy slides were initially scanned at 5× (PreScan), global images were filtered using Gaussian blur and tissue regions were identified based on Hoechst 33342 signal. The geometric centre of the tissue was then calculated and expanded to create a region of interest covering most of the tissue. Following morphological properties calculations, 50 independent non-overlapping imaging fields were placed into the tissue region of interest and subsequently imaged (ReScan) at high resolution using a confocal 63× 1.15NA water objective (see Supplementary Methods and Supplementary Fig. 10-11). For each imaging field, Z-stacks were generated at a step size of 0.5µm (100 planes in total) across all imaging channels (i.e. 405 nm, 488 nm and 555 nm wavelengths).

Cell lines were imaged in filtered 1×PBS as mounting medium (200 µl/well) using a confocal 40× 1.1NA water objective. Images from five independent non-overlapping imaging fields were acquired from each well (see Fig. 5a). For each imaging field, Z-stack images were collected at a step size of 0.5 µm (28 planes in total) across all imaging channels (i.e. 405 nm, 488 nm, 555 nm and 647 nm wavelengths).

## Image Analysis

Image analyses were performed using the Harmony software. Image analysis scripts were optimised for different sample types and immunofluorescent stains. In brief, images were reconstructed as maximum projections (collapsed Z-stacks) using basic brightfield correction. Image analyses included segmentation of nuclei, removal of border objects to only include whole cells, identification of cytoplasm and regions of interest, and subsequent centrosome detection and spot counting (see Supplementary Methods and Supplementary Fig. 12 for a detailed description). Each image analysis step included a series of quality controls based on morphological and intensity properties of identified objects. The centrosome amplification score (CA score) was estimated as the total number of centrosomes divided by the total

number of nuclei detected in each imaging field. To mitigate the effects of immunofluorescent staining background noise, auto-fluorescence and possible false positive centrosome detection, imaging fields with CA scores <0.1 were removed from downstream analysis (see Supplementary Methods and Supplementary Fig. 13 for more information and quality control cut-off estimation).

## Statistical modelling of centrosome amplification score (CA score) variabilities

To model the distribution of CA scores across imaging fields within tissue samples and within tissue types (displayed in Fig. 2), we developed a linear mixed model[59] allowing:

1. the detection of differences in average CA scores between tissue types by means of fixed effects,
2. to take take into account the CA within-tissue dependence by means of normally distributed random intercepts,
3. the spread (variance) of within-tissue CA scores to vary between tissues by means of gamma distributed standard deviations and to vary between tissue types by means of fixed effects,
4. the mean and variance of within-tissue CA scores to be dependent by using a Gaussian copula to model the association between the random intercepts of point 2 and random standard deviations of point 3,
5. the model residuals to be heavy-tailed by considering the generalised T (taking the Gaussian distribution as limit case) as conditional error distribution (i.e., distribution of the error terms given the fixed and random effects described in points 1 to 4).

Parameter estimates and corresponding inference were obtained by means of the iterative bootstrap[60] and the indirect inference method[61], respectively. Monte Carlo simulations showed that the estimator was consistent and that the confidence interval coverages were close to the 0.95 nominal values.

## H&E purity estimation

H&E sections from FFPE tissues were sent to our pathologist for tumour marking and purity estimation. In addition, Haematoxylin and Eosin (H&E) sections were scanned and subjected to HALO, an image analysis platform for quantitative tissue analysis in digital pathology. HALO's random forest classifier was used to separate the H&E image into tumour, stroma and microscope glass slide, allowing tumour purity estimation.

## DNA and RNA extraction

**FFPE tissue samples.** For each FFPE sample, multiple sections at 10 μm thickness were cut depending on tissue size and tumour cellularity assessed by a pathologist, who marked tumour areas on separate H&E stained sections to guide microdissection for DNA extraction. Marked tumour areas from unstained FFPE sections were scraped off manually using a scalpel, dewaxed in xylene and subsequently washed with 100% ethanol. Following complete removal and evaporation of residual ethanol (10 mins at 30°C), DNA was extracted using the AllPrep DNA/RNA FFPE Kit (Qiagen). DNA was eluted in 40 μl Elution buffer.

**Cell Lines.** Cell pellets of approximately 1×10⁶ cells were generated from cultured cells for each cell line outlined above and stored at −80°C until further use. DNA and RNA were extracted from cell pellets using the Maxwell® RSC Cultured Cells DNA and RNA Kit (Promega, AS1620 and AS1390, respectively) with the Maxwell® RSC 48 Instrument (Promega, AS8500).

## DNA sequencing

**Tagged-Amplicon Sequencing (TAmSeq).** Extracted DNA samples were diluted to a final concentration of 10 ng/ml using PCR certified water. Tagged-Amplicon deep sequencing was performed as previously described[62]. In short, libraries were prepared in 48.48 Juno Access Array Integrated Fluidic Circuits chips (Fluidigm, PN 101-1926) on the IFC Controller AX instrument (Fluidigm) using primers designed to assess small indels and single nucleotide variants across the coding region of the *TP53* gene. Following target-specific amplification and unique sample indexing, libraries were pooled and subsequently purified using AMPure XP beads. Quality and quantity of the pooled library were assessed using a D1000 genomic DNA ScreenTape (Agilent 5067-5582) on the Agilent 4200 TapeStation System (G2991AA), before submitting the library for sequencing to the CRUK CI Genomics Core Facility using 150 bp paired-end mode on either the NovaSeq 6000 (SP flowcell) or HiSeq 4000 system. Sequencing reads were aligned to the 1000 Genomes Project GRCh37-derived reference genome (i.e. hs37d5) using the BWA-MEM aligner. OV04 tissue and cell line data analysis and variant calling was performed as previously described[62,63]. TAmSeq data and variant calls for the BriTROC study subset was obtained from Smith et al.[58].

**Shallow Whole Genome Sequencing (sWGS).** DNA extractions were performed as described above, and quantified using Qubit quantification (ThermoFisher, Q328851). DNA samples were diluted to 75 ng in 15 μl of PCR certified water and sheared by sonication with a target of 200–250 bp using the LE220-plus Focused-Ultrasonicator (Covaris) with the following settings: 120 sec at room temperature; 30% duty factor; 180 W peak incident power; 50 cycles per burst. Sequencing libraries were prepared using the SMARTer Thruplex DNA-Seq kit (Takara), with each FFPE sample undergoing 7 PCR cycles, and all other samples undergoing 5 PCR cycles for unique sample indexing and library amplification. AMPure XP beads were used following manufacturer's recommendations to clean prepared libraries, which were subsequently eluted in 20 μl TE buffer. sWGS libraries were quantified and quality-checked using D5000 genomic DNA ScreenTapes (Agilent 5067-5588) on the Agilent 4200 TapeStation System (G2991AA) before pooling the uniquely indexed samples in equimolar ratios. Pooled libraries were sequenced at low coverage (-0.4 × coverage) on either NovaSeq 6000 S1 flowcells with paired-end 50 bp reads for clinical tissue samples, or the HiSeq 4000 with single 50 bp reads, at the CRUK CI Genomic Core Facility. Resulting sequencing reads were aligned to the 1000 Genomes Project GRCh37-derived reference genome (i.e. hs37d5) using the BWA aligner (v.0.07.17) with default parameters.

## Copy number analyses

Relative copy number data for OV04 tissue samples and ovarian cancer cell lines was obtained using the QDNAseq[64] R package (v1.24.0) to count reads within 30 kb bins, followed by read count correction for sequence mappability and GC content, and copy number segmentation. QDNAseq data were then subjected to downstream analyses using the Rascal R package[41] for ploidy and cellularity estimation and absolute copy number fitting. Absolute copy number data for BriTROC tissue samples was obtained from Smith et al. (2023) utilising comparable analysis approaches[58].

In all data sets, putative driver amplifications were detected and identified using the Catalogue Of Somatic Mutations In Cancer (COSMIC; https://cancer.sanger.ac.uk/cosmic/help/cnv/overview) definitions and thresholds for high level amplifications and homozygous deletions: *Amplification* – average genome ploidy ≤2.7 and total copy number ≥5; or average genome ploidy >2.7 and total copy number ≥9. *Deletion* – average genome ploidy ≤2.7 and total copy number = 0; or average genome ploidy >2.7 and total copy number <(average genome ploidy 2.7). In contrast, copy number gains and losses were defined as copy number > ploidy + 1, or copy number <ploidy − 1, respectively, and are not defined as distinct copy number alterations in the same manner as amplifications and deletions.

## Copy number signatures (CNS)

Copy number signatures were estimated as previously described[8] from absolute copy number data, following computation of the distributions of six genomic features: I. the breakpoint count per 10 Mb, II. the copy number of each segment, III. the copy number change points (i.e. the difference in copy number between adjacent segments), IV. the breakpoint count per chromosome arm, V. the lengths of chains of oscillating copy number segments, and VI. the size of segments.

## Fixed effect model (for CNS analyses)

To account for the compositional nature of copy number signatures (the sum of copy number signature exposures within each samples = 1), we performed additive log ratio (ALR) transformation[65] in a sample-wise manner, choosing signature 7 as the denominator. Subsequently a fixed effect regression model was applied to investigate differences in signature abundances between CA and MN high vs. low samples. For determining global statistically significant shifts in copy number signature abundances we used a Wald test.

## tMAD scores

Trimmed median absolute deviation from CN neutrality (tMAD) scores were estimated as a genome-wide summary measure of genomic abnormality as described before[45]. The R code was adjusted from https://github.com/sdchandra/tMAD. Samples were downsampled to $3 \times 10^6$ reads and allocated into 500 kb non-overlapping bins as recommended by the authors. The diploid normal ovarian surface epithelium cell line, IOSE4, which was the cell line with the least (i.e. no detectable) copy number aberrations, was chosen as normal/control sample for normalisation.

## TCGA dataset analysis

Publicly available clinical data and RNAseq V2 data from The Cancer Genome Atlas (TCGA; https://www.cancer.gov/about-nci/organization/ccg/research/structural-genomics/tcga) were downloaded using Fire-browse (http://firebrowse.org/). Centrosome amplification gene expression (CA20) signatures were calculated as previously described[30]. In brief, gene-level read counts were quantile normalised using Voom[66] and the CA20 score was estimated for each sample as the sum of the log2 median-centred gene expression levels of the CA20 signature genes: *AURKA, CCNA2, CCND1, CCNE2, CDK1, CEP63, CEP152, E2F1, E2F2, LMO4, MDM2, MYCN, NDRG1, NEK2, PIN1, PLK1, PLK4, SASS6, STIL,* and *TUBG1*. The CIN25 gene expression signature, as previously defined[33], was estimated in the same manner using the following genes: *C20orf24/TGIF2, CCNB2, CCT5, CDC45, CDC2, ESPL1, FEN1, FOXM1, H2AFZ, KIF20A, MAD2L1, MCM2, MCM7, MELK, NCAPD2, PCNA, PRC1, RAD51AP1, RFC4, RNASEH2A, TOP2A, TPX2, TRIP13, TTK, UBE2C*.

## RNA sequencing

Following RNA extractions, RNA integrity number (RIN) values were estimated using the Bioanalyzer system (Agilent). RIN values ranged from 8.3–10 with a mean of 9.7. Library preparation and RNA sequencing (RNAseq) were performed by the CRUK CI Genomics Core Facility: In brief libraries were prepared using the Illumina TruSeq standard mRNA kit and sequenced using 50 bp paired-end mode on the NovaSeq 6000 sequencer (S1 flowcell).

## RNAseq analyses

Sequencing reads were mapped to the GRCh38 (hg38) reference genome using STAR aligner v2.7.6a (https://github.com/alexdobin/STAR/releases). Following standard quality control (MultiQC[67]), raw sequencing reads were quantified against the same reference genome (GRCh38) using Salmon v1.4.0[68], https://combine-lab.github.io/salmon/). Quantified transcripts (gene-level) were imported into R using the tximport[69] in preparation for downstream analyses. Low-abundance genes were filtered out, keeping only genes with at least 25 reads in at least two samples. Gene count normalisation and differential gene expression analyses were performed using DESeq2 v1.26.0[70]. Genes with $p$ value < 0.05, FDR < 0.1 and a fold-change of at least 1.5 were considered as differentially expressed. Likelihood ratio tests were used to assess which model (simple, additive or interactive) was most appropriate for each comparison. Data was visualised using the lfcShrink() function (DESeq2 package) and the ComplexHeatmap R package v2.2.0[71]. Gene set enrichment analyses (GSEA) were performed using the fgsea R package v1.12.0 (https://github.com/ctlab/fgsea). Genes were ranked by statistical significance and fold change; and 50 hallmark sets from the human C5 MSigDB collection were used for running GSEA. 1,000 permutations were used to calculate p-values. In addition, the clusterProfiler R package v3.14.3[72] was used for gene ontology-based overrepresentation analyses.

## Drug sensitivity assays – GR metrics

Cell lines were seeded into opaque-walled 384 well plates compatible with the CLARIOstar® Plus (BMG Labtech) luminometer using a CombiDispenser (Thermo ScientificTM). For each cell line, two plates were generated–a control plate to correct for the number of divisions taking place over the course of the assay, and a test plate. 24 hours after seeding, 10 µl of CellTiter-Glo® (Promega) were added to the control plate using the CombiDispenser (Thermo ScientificTM), and after an incubation time of 30 mins the plate was read using the CLARIOstar® Plus luminometer (BMG Labtech). At the same time (24 hours post-seeding), drugs were added to the test plate using a D300e Digital Dispenser (Tecan). Plate layouts were randomised, normalised for DMSO volumes across wells, and drugs were dispensed as 10-point half-log dilution series with a DMSO concentration of 0.01%. The following compounds were used: oxaliplatin (Selleckchem S1224), paclitaxel (Selleckchem S1150), tozasertib (Selleckchem S1048), alisertib (Selleckchem S1133), barasertib (Selleckchem S1147), cw069 (Selleckchem S7336), bi2536 (Selleckchem S1109), volasertib (Selleckchem S2235), cfi400945 (Selleckchem S7552), az3146 (Selleckchem S2731), and gsk923295 (Selleckchem S7090). The highest drug concentration used for paclitaxel was 0.1 µM. For all other compounds, the highest tested concentration was 100 µM. Drug screens were performed in triplicates. Following the addition of drugs, treatment plates were incubated for 72 hours, after which cell viabilities were assessed as described above. To estimate drug sensitivity, as well as drug potency and efficacy for each cell line and compound, the growth rate inhibition metrics were estimated as previously described[47,48] using the GRmetrics R package v1.12.2 (https://bioconductor.org/packages/release/bioc/html/GRmetrics.html).

## Reporting summary

Further information on research design is available in the Nature Portfolio Reporting Summary linked to this article.

# Data availability

The raw shallow whole genome sequencing (sWGS) data and RNA sequencing data for all cell lines used in this study have been deposited to the European Nucleotide Archive (ENA) with study/project accession number PRJEB60280 and is publicly accessible for download. The raw sWGS data for OV04 tissue samples has been deposited to the European Phenome-Genome Archive (EGA) with accession number EGAD00001008121. Absolute copy number data for BriTROC-1 cases were provided by Smith et al. [58]. All raw genomic data relating to the BriTROC-1 study are available via EGA under accession code EGAS00001007292. The raw genomic sequencing data for OV04 and BriTROC-1 patient samples are available under restricted access due to patient confidentiality. Access to the OV04 and BriTROC-1 sequencing data can be obtained by authorised researchers or clinicians by applying to the relevant Data Access Committees, which aim to respond to any data access applications within a week of receiving the

request. There are no restrictions on the duration of access once granted. More information on EGA and on how to access data from EGA can be found here (https://www.ebi.ac.uk/training/online/courses/ega-quick-tour/accessing-the-data-in-the-ega/). Publicly available clinical data and RNAseq V2 data from The Cancer Genome Atlas (TCGA; https://www.cancer.gov/about-nci/organization/ccg/research/structural-genomics/tcga) were downloaded using Firebrowse (http://firebrowse.org/). The remaining data are available within the Article, Supplementary Information or Source Data files, and on GitHub (https://github.com/cmsauer/CAOV2023). Source data are provided with this paper.

## Code availability

All code required to reproduce figures has been deposited on GitHub (https://github.com/cmsauer/CAOV2023).

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

## Acknowledgements

We thank all patients who participated in and donated tissue samples to this study. The Addenbrookes Human Research Tissue Bank is supported by the NIHR Cambridge Biomedical Research Centre. We also thank the OV04 study team for their help with clinical tissue samples. We would like to thank the Cancer Research UK Cambridge Institute Microscopy, Compliance & Biobanking, Genomics, Bioinformatics, and IT & Scientific Computing core facilities for their support with various aspects of this study. We would like to thank D. Provencher and A.-M Mes-Masson for kindly donating a subset of 25 ovarian cancer cell lines to us that were included in this study. These cell lines were derived with the support of the Banque de tissus et de données of the Réseau de recherche sur le cancer of the Fonds de recherche du Québec - Santé (FRQS) affiliated with the Canadian Tumor Repository Network (CTRNet). We also thank Fanni Gergely for her kind donation of the CDK5RAP2 antibody, and Gayathri Chandrasekaran for her help and advice on the centrosome staining protocol. We acknowledge funding and support from Cancer Research UK, and the Cancer Research UK Cambridge Centre: A22905 (C.M.S., J.B.D.); A25177 (M.A.V.R), A25117 (K.H.). L.M.G. was supported by the Wellcome Trust PhD programme in Mathematical Genomics and Medicine (grant number RG92770). F.C.M. was funded by the Experimental Medicine Initiative from the University of Cambridge, the Academy of Medical Sciences (SGL016_1084), Cancer Research UK (C53876/A24267 and A25117) and the Rosetrees and Stoneygate Trusts (A2854). This research was also supported by the Ovarian Cancer Action (grant number 006; I.A.M.), and the NIHR Cambridge Biomedical Research Centre (BRC-1215-20014). Work in the Cancer Molecular Diagnostics Laboratory/Blood Processing Laboratory was supported by the NIHR Cambridge Biomedical Research Centre, Cancer Research UK Cambridge Centre and the Mark Foundation Institute for Integrated Cancer Medicine. The views expressed are those of the authors and not necessarily those of Cancer Research UK, the NIHR or the Department of Health and Social Care. The funders had no role in study design, data collection and analysis, decision to publish, or preparation of the manuscript.

## Author contributions

Conceptualization, C.M.S., J.T., M.V. and J.D.B.; Methodology, C.M.S. and D.L.C; Validation, C.M.S.; Formal Analysis, C.M.S. and D.L.C.;

Investigation, C.M.S., J.A.H., A.M.P., J.G., J.T., M.V.; Resources, C.M.S., D.L.C., T.B., A.M.P., A.S., M.D.E., P.S., K.H., M.A.V.R., L.M.G, A-M.M-M., D.E., D.M., A.H., I.A.M., M.J.L., F.C.M., and J.T.; Data Curation, C.M.S., D.L.C., T.B., P.S., and K.H.; Writing - Original Draft, C.M.S.; Writing - Review & Editing, C.M.S., J.T., M.V., and J.D.B.; Visualisation, C.M.S.; Supervision, C.M.S., J.T., M.V., and J.D.B.; Project Administration, C.M.S. and M.V. Funding Acquisition, J.D.B.

## Competing interests

J.D.B. is a co-founder and shareholder of Tailor Bio Ltd. There are no further conflicts of interests to declare.
