## [Peer Review File · Nature Communications]

Molecular landscape and functional characterization of centrosome amplification in ovarian cancerReviewers' Comments:

Reviewer #1:

Remarks to the Author:

This in-depth study by Sauer et al sheds significant new light on the occurrence of centrosome amplification (CA) in ovarian cancer and its use as a clinical prognostic marker. The authors develop a novel, high-throughput microscopy-based analysis pipeline for detecting centrosomes in an automated manner. Employing this on a large number of FFPE tissue samples, they reveal that CA is a common phenotype in HGSOC tissue, yet significant inter- and intra-tumour heterogeneity exists. The authors then repeat this analysis in a panel of 73 cell lines, validating that CA is highly prevalent in ovarian cancer, and allowing the follow on analyses of genomic alteration patterns, gene expression studies, and some functional testing. Interestingly, they present evidence that cell lines with higher rates of CA may be less sensitive to chemotherapeutic agents, including paclitaxel, although this appears only to have been seen in the cell lines, since no clear association between clinical features and CA levels was observed upon analysis of CA in FFPE tissues.

The manuscript is extremely well written and presents a very large amount of carefully collected data that will be of significant interest to the fields of ovarian cancer, centrosome abnormalities in cancer, and genomic instability in cancer. The authors analyse a commendable number of both tissue samples and cell lines, which provides confidence in the data and their overall conclusions. Characterisations of centrosome abnormalities have been previously performed in ovarian cancer, but never to this scale, meaning this study adds significant new knowledge to the field. I am therefore very supportive of this study being published in Nature Communications. Nevertheless, there are a few aspects which could be improved upon:

Major points:

1. Can they expand further upon why their approach is an advance over the recent study of centrosome abnormalities in HGSOC from the Basto Lab approach, and can they reconcile differences in findings (namely the discovery of loss of centrosomes in the aforementioned study)?
2. Can the authors include their thoughts in the Discussion about why, despite the identified resistance to paclitaxel seen in high CA cell lines, no clear association between CA and clinical features could be observed from the FFPE analysis?
3. Fig 5B – it was surprising to me that the three normal cell lines used in this study (FT194, IOSE4 and FT246) have relatively high CA scores. Indeed, ~40% of non-mitotic FT246 cells were found to have 2 or more centrosomes. The authors speculate that the reason for this could be that transformation and/or immortalisation of normal cells induces CA. However, a previous analysis found relatively low rates of centrosome amplification in h-TERT-immortalised fallopian tube serous epithelial cell lines (<https://doi.org/10.1158/0008-5472.CAN-19-0852>). These normal controls also displayed low rates of CIN in comparison to HGSOC lines. Is there any way to test the authors' explanation for their observations? Or perhaps a greater selection of normal cell lines could be included in the study? As a minimum, the authors should be more careful when making a general statement regarding the lack of suitability of immortalised normal cells for CA/CIN studies (lines 235-237).
4. Using a CA cutoff of less than 2 (1.83) might mean that simply having a higher percentage of cycling cells would be more likely to lead to the calling of a given sample as exhibiting CA, simply because more cells would contain 2 centrosomes. Can they control for this by determining the centrosome number in highly proliferative tissue, or correcting using the proportion of cells in S/G2, or mitosis for example? Or can they show that the majority of cells are in G1 (where one centrosome would be expected)?
5. Figure 6: Again, using a cut off of 2 or more centrosomes = CA means that simply having a higher proliferative rate could result in a false positive for CA. This might explain why the normal cells also show unexpectedly high rates of 'CA'. Indeed, they observe a correlation between CA and mitotic index (6c), suggesting this could be a confounding factor. Can they use a different cut off (3 or more

centrosomes), or control for proliferation differences between (i) the different cell lines and (ii) the different oxygen concentration conditions? Alternatively, providing the raw data (centrosome number per cell) might give a better idea of the true range of CA.

6. Fig 6C – RE: the correlation between CA score and mitotic index. The authors interpret this as cells with CA being delayed in mitosis (line 388) yet do not attempt to confirm this by e.g. analysing the length of time to complete mitosis from NEBD. Performing this in even a small subset of high CA and low CA cell lines would strengthen the conclusions drawn.

7. Fig 6D/E – despite not observing an increase in CA in HGSOC compared to other OvCa subtypes, the authors do see an increase in MN frequency. Does this suggest that mechanisms other than CA are perhaps more important in driving CIN in HGSOC (e.g. replication stress, mitotic checkpoint defects etc.) and that CA makes only a relatively minor contribution?

Minor points:

1. Fig 1B,C – could the DAPI, centrosomes, and identified centrosomes colours be indicated on the images, as this would make this figure easier to interpret?

2. Line 205: 'pHH3' is used, but could the full name be given instead as it is not clear what this means.

3. Line 208 (and 225): 'non-mitotic cells' – does this mean interphase cells, or cells that are not cycling?

4. Line 227: what does 'non-pertubated ovarian cancer cell lines' mean?

5. Line 129-131 the description of the hierarchical statistic model isn't very clear

6. Fig 1E – the order in which sample types are listed in the legend do not match the graph.

7. Fig 2A – the colour of the liver samples BriTROC and OV04 are quite similar, possibly more distinguishable colours would make this figure easier to understand.

8. Fig 3C - add 'Red stars indicate CA score medians for patient with multiple tissue samples' to the key to remove any confusion.

9. Fig 3E – is there heterogeneity in centrosome numbers in the liver samples? Would assume no, but would be good to see this on the figure as a control.

10. Fig 4A – when comparing CA scores from tissue samples between OvCa subtypes, the authors only compare HGSOC to endometrioid. Is data available for other subtypes (e.g. LGSOC, clear cell, mucinous)?

11. Fig 5B – could the authors clarify the approach taken to calculate CA scores for the cell lines? If it was indeed $(\sum \text{Centrosomes} / \sum \text{Nuclei})$, I fail to see how this could result in scores close to zero for e.g. PEO6 cells, unless centrosomes were unable to be detected in some cell lines.

12. It may be informative to include a plot similar to Fig 4B whereby the mean CA scores from cell lines are compared by disease stage (if this information is available for the cell lines used).

13. It is interesting that the authors do not observe any difference in CA frequencies across different subtypes from the cell line data (Fig 6D), particularly given the significant difference observed between HGSOC and endometrioid tissue samples earlier in the paper (Fig 4A). The authors should comment on this inconsistency and speculate on the reasoning for this.

Reviewer #2:

Remarks to the Author:

Sauer et al address an interesting question of centrosome amplification in ovarian cancer. They developed and used a high-throughput microscopy on 287 clinical HGSOC tumour tissues and 73 ovarian cancer cell lines and show that CA through centriole overduplication is a highly recurrent and heterogeneous feature of HGSOC and is strongly associated with CIN and genome subclonality. Using cell-based studies they show that high CA is associated with increased multi-treatment resistance particularly to paclitaxel, bringing new biological insights into CA and its role in ovarian cancer evolution and treatment resistance.

The manuscript is well written, and results are convincing and clearly presented. They have also contributed by developing an automated, high-throughput imaging-based assay for CA assessment in clinical samples and cell lines.

I have a few comments that could be addressed to improve the value of the manuscript

Did the authors define the CA only in the cancer cells of the clinical samples or were the data quantified on all cells in the tissues, including "normal" stromal and immune cells? I would suggest to explore the possibility to separate the CA signal coming from the cancer cells only - using e.g. morphology to separate cancer cells from normal cells in the images. Perhaps the results might reveal additional correlations to treatment responses and outcomes when they are not confounded by the signal coming from normal cells. Did the authors compare the effect of tumor purity on the CA values in the clinical samples? The finding on the increased heterogeneity of post -neoadjuvant treatment samples would support the notion that this heterogeneity can be solely attributed to the purity of tumor cells over normal cells as the tumor purity is known to be much lower in the post neoadjuvant samples.

Minor comments:

Row 206: These analyses confirmed that ovarian cancer cell lines recapitulate the high prevalence of CA observed in HGSOC tissue samples with a mean CA frequency of 26.2% (range 0.5–50.4%; non-mitotic cells with two or more centrosomes; Fig. 6a, b)."

This phrase is confusing. Consider adding a new line describing Fig. 6a and Fig 6b.

Row 269 "Consistent with observation that CA and MN are moderately correlated (Spearman's $R = 0.38$, $p = 0.0015$; Fig. 7c)"

The previous sentence does not correspond to Fig. 7c, is it perhaps Fig. 6c?

Point-to-point Response to Reviewers' comments**Reviewer #1 – expertise in chromosome biology and instability****Remarks to the Author:**

This in-depth study by Sauer et al sheds significant new light on the occurrence of centrosome amplification (CA) in ovarian cancer and its use as a clinical prognostic marker. The authors develop a novel, high-throughput microscopy-based analysis pipeline for detecting centrosomes in an automated manner. Employing this on a large number of FFPE tissue samples, they reveal that CA is a common phenotype in HGSOc tissue, yet significant inter- and intra-tumour heterogeneity exists. The authors then repeat this analysis in a panel of 73 cell lines, validating that CA is highly prevalent in ovarian cancer, and allowing the follow on analyses of genomic alteration patterns, gene expression studies, and some functional testing. Interestingly, they present evidence that cell lines with higher rates of CA may be less sensitive to chemotherapeutic agents, including paclitaxel, although this appears only to have been seen in the cell lines, since no clear association between clinical features and CA levels was observed upon analysis of CA in FFPE tissues.

The manuscript is extremely well written and presents a very large amount of carefully collected data that will be of significant interest to the fields of ovarian cancer, centrosome abnormalities in cancer, and genomic instability in cancer. The authors analyse a commendable number of both tissue samples and cell lines, which provides confidence in the data and their overall conclusions. Characterisations of centrosome abnormalities have been previously performed in ovarian cancer, but never to this scale, meaning this study adds significant new knowledge to the field. I am therefore very supportive of this study being published in Nature Communications. Nevertheless, there are a few aspects which could be improved upon:

We thank the reviewer for this very positive assessment of our work and the detailed comments below.

Major points:

1. Can they expand further upon why their approach is an advance over the recent study of centrosome abnormalities in HGSOc from the Basto Lab approach, and can they reconcile differences in findings (namely the discovery of loss of centrosomes in the aforementioned study)?

Please refer to Table 1 and 2 below summarizing the key differences and similarities between our current study and Morretton et al. (2022) (Basto Lab). In brief, Morretton et al. (2022) describe centrosome loss as a recurrent phenotype in ovarian cancers and further follow up on these observations utilising spheroid models. In their study a total of 100 ovarian cancer tissues (20µm fresh-frozen) sections were stained with PCNT and CDK5RAP2 for centrosome visualisation and detection, and analysis was performed manually counting spots in up to 10 imaging fields per tissue. This manual curation and analysis of centrosome allowed annotation of individual tissue regions within imaging fields that displayed a low centrosome-nucleus

index (i.e. centrosome loss). In comparison, our study focussed on the large scale phenotypic and molecular characterisation of centrosome amplification in 287 ovarian cancer tissues (+31 normal control tissues) and 73 cell line models. For our tissue cohort, 25µm FFPE sections were stained, also using PCNT and CDK5RAP2, and up to 50 independent imaging fields were scanned and analysed. Importantly, image analysis was performed in an automated manner, focussing on the tissue wide characterisation of the centrosome amplification phenotype. Given the limitations of FFPE staining and challenges related to imaging thick tissue sections with vastly disorganised tissue structures (as is the case in HGSOC), we chose not to further explore regions of potential centrosome loss, as we would not be able to confidently differentiate between true centrosome loss and tissue sectioning bias resulting in lower counts of centrosomes.

Nevertheless, the phenotypes of centrosome loss (in sub-regions of tissues) and overall tissue-wide centrosome amplification are not mutually exclusive as outlined in the Peer Review document for Morretton et al (2023) paper and the previous bioRxiv version of this publication: <https://www.biorxiv.org/content/10.1101/623983v2>.

Interestingly, Morretton et al., like us, observe high prevalence of centrosome amplification (~60%) in ovarian cancer tissue samples (bioRxiv Fig. 2E, Fig. 3B, and Peer Review Document). This is remarkably similar to the prevalence of 63.5% reported by us in the OV04 and BriTROC cohorts combined. In addition, both studies highlight marked tissue heterogeneity in regard to the centrosome amplification phenotype, which further highlights the compatibility of both regional centrosome loss and tissue-wide centrosome amplification.

DIFFERENCES	Morretton et al. (2022)	Sauer et al. (2023)
Main study focus	Description and functional characterisation of centrosome loss in ovarian cancer tissues and induced spheroid models	Large scale characterisation of centrosome amplification in ovarian cancer tissues and very extensive cell line collection and association with molecular features
Cohort size	100 ovarian cancer samples and ~20 normal control tissues samples	287 ovarian cancer and 31 normal control tissue samples
Image Analysis	Manual spot counting and image analysis, with focus on curated individual tissue regions	Automated large scale image analysis at tissue-wide scale
Tissue Type	Fresh frozen (FF); 20µm sections	Formalin-fixed paraffin embedded (FFPE); 25µm sections

Table 1. Main differences between Morretton et al. (2022) *EMBO Mol Med* and our current study (Sauer et al. 2023).

SIMILARITIES	Morretton et al. (2022) & Sauer et al. (2023)
Centrosome staining	Both used PCNT and CDK5RAP2 markers for centrosome staining and identification
Observed prevalence of centrosome amplification (CA)	Both observe and describe a prevalence of ~60% of CA in ovarian cancer tissues. (Morretton et al: see https://www.biorxiv.org/content/10.1101/623983v2.full Figure 2E and Fig 3B; and Peer Review correspondence of the EMBO Mol Med publication; Sauer et al: combined 63.5% of tissues show CA). Both describe heterogeneity of this phenotype across ovarian cancer tissues.

Table 2. Main similarities shared between Morretton et al. (2022) *EMBO Mol Med* and our current study (Sauer et al. 2023).

2. Can the authors include their thoughts in the Discussion about why, despite the identified resistance to paclitaxel seen in high CA cell lines, no clear association between CA and clinical features could be observed from the FFPE analysis?

These comparisons are complex as patients are treated with carboplatin and paclitaxel which confounds the analysis of the predictive power of CA for paclitaxel response. In addition, there are other factors that can influence treatment response and disease outcome including stage, residual disease after debulking surgery, HRD status and age. We do not have clinical response data (i.e. RECIST measurements) so can only investigate the survival outcome (which may also be confounded by other factors).

Given this complexity we are likely underpowered to observe moderate associations between CA and clinical disease outcome. For example, we are not observing a clear effect for age in the OV04 study group, but that of course does not mean that patient age is not a prognostic hazard. More detailed and focused follow-up studies will be required to further investigate the link between CA and paclitaxel response.

3. Fig 5B – it was surprising to me that the three normal cell lines used in this study (FT194, IOSE4 and FT246) have relatively high CA scores. Indeed, ~40% of non-mitotic FT246 cells were found to have 2 or more centrosomes. The authors speculate that the reason for this could be that transformation and/or immortalisation of normal cells induces CA. However, a previous analysis found relatively low rates of centrosome amplification in h-TERT-immortalised fallopian tube serous epithelial cell lines (<https://doi.org/10.1158/0008-5472.CAN-19-0852>). These normal controls also displayed low rates of CIN in comparison to HGSOC lines. Is there any way to test the authors' explanation for their observations? Or perhaps a greater selection of normal cell lines could be included in the study? As a minimum, the authors should be more careful when making a general statement regarding the lack of suitability of immortalised normal cells for CA/CIN studies (lines 235-237).

Thank you for this interesting comment. To address this further we have analysed copy number profiles in four non-cancerous normal cell lines (**Figure 1**). This showed that in particular FT246 (the cell line in which ~40% of non-mitotic cells showed CA) had significant copy number aberrations with gain, losses and focal amplifications, whereas the fallopian tube cell lines (FNE1 and FNE2) included in the highlighted study by Tamura et al. had low rates of CIN which might explain the differences in CA levels we observed in the Fallopian

Tube cell lines described in our present study. Please also note that Tamura et al. have analysed centrosome numbers in prometaphase (mitotic) cells, whereas our analyses have focussed on non-mitotic interphase cells.

We have now refined and expanded the statement in lines 235-237.

Revision Figure 1. Copy number profiles of non-cancerous cell lines for (a) FT33, (b) FT194, (c) FT246 and (d) IOSE4. Please note that no centrosome data is available for FT33 in the present study. Nevertheless, this cell line was included in the copy number analysis to illustrate the point that some fallopian tube cell lines do indeed display copy number aberrations and CIN.

4. Using a CA cutoff of less than 2 (1.83) might mean that simply having a higher percentage of cycling cells would be more likely to lead to the calling of a given sample as exhibiting CA, simply because more cells would contain 2 centrosomes. Can they control for this by determining the centrosome number in highly proliferative tissue, or correcting using the proportion of cells in S/G2, or mitosis for example? Or can they show that the majority of cells are in G1 (where one centrosome would be expected)?

Thank you for raising this important point. The majority of OV04 tissues are post-treatment and generally contain very few cycling cells. Similarly, tissue from pre-treatment samples were also not highly proliferative and the detection of mitotic cells in these tissues was rare. Please note that the cutoff of 1.83 does not directly translate to a cut off of 2 centrosomes per cell. The cut point was estimated based on the distribution of the numbers of detected centrosomes in control experiments with normal tissue samples and takes into account other sources of noise from staining and imaging (e.g. not all centrosomes are detectable as a centrosome might fall outside of the sectioning plane when cutting through a cell). As such the cutoff is rather conservative and based on the 95% confidence interval of the highest CA score Fallopian Tube control tissue sample (as outlined commencing at line 119). We now provide the raw tissue centrosome count data for interested readers to re-analyse using different CA cutoff thresholds (also requested below).

5. Figure 6: Again, using a cut off of 2 or more centrosomes = CA means that simply having a higher proliferative rate could result in a false positive for CA. This might explain why the normal cells also show unexpectedly high rates of 'CA'. Indeed, they observe a correlation between CA and mitotic index (6c), suggesting this could be a confounding factor. Can they use a different cut off (3 or more centrosomes), or control for proliferation differences between (i) the different cell lines and (ii) the different oxygen concentration conditions? Alternatively, providing the raw data (centrosome number per cell) might give a better idea of the true range of CA.

In contrast to the cut point used in tissue analyses, the CA cutoff of 2 or more centrosomes in cell line experiments is applied on a single-cell basis. Cells in culture were co-stained with Hoechst, cytokeratin (to define cell boundaries and to assign each detected centrosome to an individual cell) and phospho-histone H3 (phh3). As illustrated in the Figure 2 below, phh3 starts to be expressed in G2, at which point centrosome separation and maturation takes place, and is expressed throughout mitosis. Therefore, cells with any phh3 signal were strictly excluded from our centrosome analysis, which means that our cutoff or "2 or more centrosomes" is only applied to single cells which are in interphase (but not G2) and should therefore only have 1 centrosome (i.e. PCM foci). Note that although centrosomes start to duplicate in S phase, separation of centrioles does not take place until G2 phase, which means that duplicating centrosomes in S phase will still be detected as one centrosome (defined as one PCM focus) based on PCNT and CDK5RAP2 staining. We now provide the raw centrosome count data for the cell line experiments for interested readers to re-analyse.

Figure 2. Schematic adapted from Ren et al. 2018. *Cell Cycle*. DOI: 10.1080/15384101.2018.1426416, depicting BrdU and phh3 labelling/expression throughout the cell cycle.

Please also refer to Tang et al. 2012. *Biol Open*. DOI: <https://doi.org/10.1242/bio.20122659>, Figure 5 for example immunofluorescent images of phh3 staining throughout different stages of the cell cycle.

6. Fig 6C – RE: the correlation between CA score and mitotic index. The authors interpret this as cells with CA being delayed in mitosis (line 388) yet do not attempt to confirm this by e.g. analysing the length of time to complete mitosis from NEBD. Performing this in even a small subset of high CA and low CA cell lines would strengthen the conclusions drawn.

Our analysis focuses on the large-scale molecular characterisation of CA in cell lines and ovarian cancer tissues and detailed time lapse studies is beyond our experimental capabilities and scope for the present study. However, this would indeed be an interesting analysis to perform in a follow up study. We have now highlighted this in the discussion and also adjusted our interpretation in line 388.

7. Fig 6D/E – despite not observing an increase in CA in HGSOC compared to other OvCa subtypes, the authors do see an increase in MN frequency. Does this suggest that mechanisms other than CA are perhaps more important in driving CIN in HGSOC (e.g. replication stress, mitotic checkpoint defects etc.) and that CA makes only a relatively minor contribution?

This important point opens up a very interesting discussion about the role of CA in cancers, i.e. the question whether CA is a bystander vs. a driver. We do not believe that our data and analyses can firmly answer this question. However, given the high prevalence of CA in HGSOC, and correlation with increased genomic aberration, we do think that it is an important factor in driving CIN. In addition, we do believe that it is most likely a compendium of different mechanisms involved in driving CIN in HGSOC (also see Macintyre et al. *Nat Gen.* 2018, describing that HGSOC genomes are shaped by multiple mutational processes), and that in some cases CA has a stronger effect than in others depending on the genetic background and evolution of a given HGSOC tumour.

Minor points:

1. Fig 1B,C – could the DAPI, centrosomes, and identified centrosomes colours be indicated on the images, as this would make this figure easier to interpret?

This has now been added.

2. Line 205: ‘pHH3’ is used, but could the full name be given instead as it is not clear what this means.

Thank you. This has now been corrected.

3. Line 208 (and 225): ‘non-mitotic cells’ – does this mean interphase cells, or cells that are not cycling?

We are here referring to pph3-negative interphase cells, i.e. cells which are not currently in G2 or undergoing mitosis. This has now been clarified in the main text.

4. Line 227: what does ‘non-pertubated ovarian cancer cell lines mean?

This phrase has now been removed to avoid confusion. With this term we were referring to “native” CA (as opposed to experimentally induced CA) in ovarian cancer cell lines, since most studies to date showed CA associations with MN in settings where CA was either induced or inhibited experimentally.

5. Line 129-131 the description of the hierarchical statistic model isn’t very clear

Thank you, we have now simplified this in the main text. Note that detailed information on the model is detailed in the methods section.

6. Fig 1E – the order in which sample types are listed in the legend do not match the graph.

This has now been updated.

7. Fig 2A – the colour of the liver samples BriTROC and OV04 are quite similar, possibly more distinguishable colours would make this figure easier to understand.

This has now been updated. Colours have also been changed accordingly in Figures 1d-e, 3c-e and 4a-e.

8. Fig 3C - add 'Red stars indicate CA score medians for patient with multiple tissue samples' to the key to remove any confusion.

This has now been updated.

9. Fig 3E – is there heterogeneity in centrosome numbers in the liver samples? Would assume no, but would be good to see this on the figure as a control.

We also did observe some degree of centrosome number heterogeneity among the liver samples. As such, they were not included as a control in Fig 2. We speculate that this heterogeneity might be due to differing tissue architecture and cell types within the liver. Moreover, hepatocytes with CA frequently display different degrees of CA, rather than all liver cells having precisely 4 centrosomes. The variance of centrosome amplification scores detected in different imaging fields of analysed liver tissue samples is depicted in the whiskers of the boxplots in **Fig. 2a**. Additionally, a tissue-wide representation example of the individual imaging fields across a liver tissue is shown in Supplementary Fig. 1a, illustrating the heterogeneity observed.

10. Fig 4A – when comparing CA scores from tissue samples between OvCa subtypes, the authors only compare HGSOc to endometrioid. Is data available for other subtypes (e.g. LGSOC, clear cell, mucinous)?

Our study cohort and tissue samples selected from the OV04 study were limited to HGSOc cases.

The BriTROC study recruited and enrolled ovarian high grade serous and grade 3 endometrioid carcinoma patients who had relapsed following at least one line of platinum-based chemotherapy.

As such, no other subtypes were available to this study.

11. Fig 5B – could the authors clarify the approach taken to calculate CA scores for the cell lines? If it was indeed (Σ Centrosomes / Σ Nuclei), I fail to see how this could result in scores close to zero for e.g. PEO6 cells, unless centrosomes were unable to be detected in some cell lines.

Yes, this was indeed the approach, and we do agree that there were some cell lines that had apparent centrosome loss. We have now highlighted this more clearly in the results section of the main manuscript text.

12. It may be informative to include a plot similar to Fig 4B whereby the mean CA scores from cell lines are compared by disease stage (if this information is available for the cell lines used).

This would indeed be an interesting analysis. However, unfortunately this data is not available for the vast majority of cell lines used.

13. It is interesting that the authors do not observe any difference in CA frequencies across different subtypes from the cell line data (Fig 6D), particularly given the significant difference observed between HGSOc and endometrioid tissue samples earlier in the paper

(Fig 4A). The authors should comment on this inconsistency and speculate on the reasoning for this.

These inconsistencies may be due to the following reasons:

- we are underpowered to detect differences owing to low numbers of non-HGSOC cell line subtypes
- cell lines are often misclassified (uncertainty of classification; also see Domcke et al. *Nat Commun.* 2013.)
- most cell line models have been continuously passed for >30 years and further drift in CA and CIN may have occurred
- only four endometrioid cancer samples were included in the BriTROC sub-cohort (TMA samples) for centrosome staining. More representative patient groups for different ovarian cancer subtypes will be required to confirm and further investigate differences in the CA phenotype across histological subtypes.

Reviewer #2 – expertise in ovarian cancer genomics

Remarks to the Author:

Sauer et al address an interesting question of centrosome amplification in ovarian cancer. They developed and used a high-throughput microscopy on 287 clinical HGSOC tumour tissues and 73 ovarian cancer cell lines and show that CA through centriole overduplication is a highly recurrent and heterogeneous feature of HGSOC and is strongly associated with CIN and genome subclonality. Using cell-based studies they show that high CA is associated with increased multi-treatment resistance particularly to paclitaxel, bringing new biological insights into CA and its role in ovarian cancer evolution and treatment resistance.

The manuscript is well written, and results are convincing and clearly presented. They have also contributed by developing an automated, high-throughput imaging-based assay for CA assessment in clinical samples and cell lines.

We thank the reviewer for their positive assessment of our work!

I have a few comments that could be addressed to improve the value of the manuscript.

Did the authors define the CA only in the cancer cells of the clinical samples or were the data quantified on all cells in the tissues, including “normal” stromal and immune cells? I would suggest to explore the possibility to separate the CA signal coming from the cancer cells only - using e.g. morphology to separate cancer cells from normal cells in the images. Perhaps the results might reveal additional correlations to treatment responses and outcomes when they are not confounded by the signal coming from normal cells. Did the authors compare the effect of tumor purity on the CA values in the clinical samples? The finding on the increased heterogeneity of post -neoadjuvant treatment samples would support the notion that this heterogeneity can be solely attributed to the purity of tumor cells over normal cells as the tumor purity is known to be much lower in the post neoadjuvant samples.

Thank you for these important comments. We carefully examined the possibility of tumour purity being a confounding factor. Please note that the BriTROC tissue microarrays (TMAs) were constructed using regions selected for high tumour purity , and H&E sections of resulting

TMA samples were subsequently inspected to only include tissues of high tumour purity in this analysis.

In contrast, the OV04 samples were studied as full-face sections, with imaging fields placed into the tissue sections as shown and described in the Supplementary Methods (Methods Fig. 5). If purity was a detrimental confounding factor in the OV04 tissue analysis, one would expect the CA values for the BriTROC TMA samples to be significantly higher than for the OV04 tissue samples. As illustrated in Fig. 2a, this was not the case. Furthermore, an independent study (Morretton et al. EMBO Mol Med. 2023) performed on ovarian cancer fresh frozen tissue samples reported similar CA prevalences in fresh frozen ovarian cancer tissue samples following manual curation and analysis of confocal microscopy images. Please refer to our response to Reviewer #1's comment 1 for more detailed information. This further strengthens and confirms our tissue study results, despite tumour purity being a potential confounding factor.

Please also note that CA scores were normalised to control tissues, to a) account for technical (and batch) staining limitations, and b) to penalise tissue samples of potentially low tumour purity.

Lastly, to investigate this further we analysed H&E sections of corresponding tissues taken at the time of centrosome staining. We asked a pathologist to provide cellularity estimates and performed image analysis (random forest classifier) using HALO (please refer to Sauer et al. 2022 (bioRxiv), doi: <https://doi.org/10.1101/2021.07.19.452658> for further information). As shown in **Revision Fig. 3** below, no correlation was observed between the cellularity assessment by a clinical pathologist and median CA scores (Pearson's $R=0.19$; $p=0.069$). Only a very weak correlation was observed between Cellularity estimates generated using HALO's random forest classifier and median CA scores (Pearson's $R=0.21$; $p=0.048$).

Together, this suggests that while normal/stromal contamination is a likely contaminating factor and a limitation of our approach, it does not detrimentally affect our results, and does not support the notion that our observed heterogeneity is purely due to normal contamination.

Please also refer to our response to Reviewer #1's comment 2 regarding correlations with clinical features and treatment response.

In addition, we agree that being able to directly differentiate between tumour and "normal"/stromal cells in our staining experiment would be a more ideal experimental set-up. However, using morphology for this kind of differentiation is not possible in DAPI/Hoechst-stained immunofluorescent image sections, in particular in 25 μ m thick FFPE sections with highly disorganised tissue architectures where several layers of cells overlap. We have attempted to use additional markers (e.g. Pax8 and cytokeratin) to differentiate between stromal and epithelial cancer cells, which would require staining centrosomes with only one centrosome marker. However, owing to FFPE-tissue processing limitations, staining limitations and tissue thickness these approaches were not successful. As unambiguous identification in FFPE tissue sections required two centrosome markers, we prioritized these IF markers.

Minor comments:

Row 206: These analyses confirmed that ovarian cancer cell lines recapitulate the high prevalence of CA observed in HGSOc tissue samples with a mean CA frequency of 26.2% (range 0.5–50.4%; non-mitotic cells with two or more centrosomes; Fig. 6a, b)."

This phrase is confusing. Consider adding a new line describing Fig. 6a and Fig 6b.

Thank you, this has now been addressed.

Row 269 "Consistent with observation that CA and MN are moderately correlated (Spearman's $R = 0.38$, $p = 0.0015$; Fig. 7c)"

The previous sentence does not correspond to Fig. 7c, is it perhaps Fig. 6c?

It is indeed. Thank you for pointing this out, this has now been corrected in the text.

Reviewers' Comments:

Reviewer #1:

Remarks to the Author:

I thank the authors for their careful and thoughtful responses to my comments. I am happy with their revisions, recommend publication in Nature Communications, and congratulate them on a very interesting and valuable piece of research.

Reviewer #2:

Remarks to the Author:

The authors have now answered all my questions and added new data to further support the conclusions of the study.

Point-to-point Response to Reviewers' comments

Reviewer #1 – expertise in chromosome biology and instability

Remarks to the Author:

I thank the authors for their careful and thoughtful responses to my comments. I am happy with their revisions, recommend publication in Nature Communications, and congratulate them on a very interesting and valuable piece of research.

We thank the reviewer for their feedback, positive assessment, and support of our work.

Reviewer #2 – expertise in ovarian cancer genomics

Remarks to the Author:

The authors have now answered all my questions and added new data to further support the conclusions of the study.

We thank the reviewer for their feedback, positive assessment, and support of our work.